# Basigin links altered skeletal stem cell lineage dynamics with glucocorticoid-induced bone loss and impaired angiogenesis

Thomas H. Ambrosi [1,6] ✉, David Morales[1,6], Kun Chen [1,6], Ethan J. Hunt [1], Kelly C. Weldon[1], Amber N. Maifeld[1], Fatima I. M. Chavez[1], Yuting Wang[1,2], Liming Zhao[2], Luke Wang[1], Matthew P. Murphy [2], Amin Cressman[3], Erika E. Wheeler[1,4], Augustine M. Saiz [1], J. Kent Leach [1,4], Fernando A. Fierro [3], Charles K. F. Chan [2,7] & Nancy E. Lane [5] ✉

Glucocorticoid (GC) induced osteoporosis (GIOP) and osteonecrosis remain a significant health issue with few approved therapies. Here, we investigate the cellular and molecular processes by which GCs affect osteogenesis and angiogenesis. We find that GC treatment reduces bone mass through decreased bone formation by skeletal stem cells (SSCs). Concomitantly, endothelial cells increase in number but display distorted phenotypical features. Transplantation studies of SSCs combined with molecular analysis by single cell RNA-sequencing and functional testing of primary human cells tie GC-induced skeletal changes to altered stem cell differentiation dynamics. This in turn perpetuates reduced osteogenesis and vascular malformation through direct SSC-endothelial crosstalk mediated at least in part by Basigin. The genetic deletion of Basigin in the skeletal lineage as well as antibody-mediated blockade of Basigin during GC treatment prevents bone loss. Intriguingly, when administered to 2-year-old mice, anti-Basigin therapy reinstates bone remodeling to significantly improve bone mass. These findings provide therapeutic vantage points for GIOP and potentially other conditions associated with bone loss.

Glucocorticoids (GCs) are potent anti-inflammatory compounds; however, continued exposure results in bone loss and osteonecrosis[1]. Since skeletal stem cells (SSCs) are crucial for maintaining skeletal homeostasis, it is tempting to speculate that GCs might affect their function. Recently, detrimental changes to angiogenesis have also been implicated in GC-induced bone loss[2,3]. Nevertheless, the mechanism by which GCs alter vascularity remains unknown. Our study aimed to evaluate GC effects on both osteogenesis and angiogenesis, as well as whether treatment with parathyroid hormone (hPTH 1-34) modifies the effects in a mouse model of GC-induced bone loss.

[1]Department of Orthopaedic Surgery, University of California at Davis Medical School, Sacramento, CA, USA. [2]Institute for Stem Cell Biology and Regenerative Medicine, Stanford University School of Medicine, Stanford, CA, USA. [3]Institute for Regenerative Cures, University of California Davis, Sacramento, CA, USA. [4]Department of Biomedical Engineering, University of California at Davis, Davis, CA, USA. [5]Department of Medicine, Division of Rheumatology, University of California Davis, Sacramento, CA, USA. [6]These authors contributed equally: Thomas H. Ambrosi, David Morales, Kun Chen. [7]Deceased: Charles K. F. Chan. ✉e-mail: thambrosi@health.ucdavis.edu; nelane@ucdavis.edu

GCs are frequently prescribed for both acute and chronic medical conditions to reduce inflammation and immune system activity. GCs are known to alter bone remodeling by reducing osteoblast activity, stimulating osteoclast maturation and activity and reducing gonadal hormone production, which indirectly activates osteoclastogenesis[1]. Bone loss induced by GCs tends to have two phases, with an initial rapid loss of bone mass, particularly trabecular, followed by a sustained, gradual reduction that results in the loss of cortical bone. Independent of the GC-induced loss of bone mass, subjects treated with GCs often rapidly reduce the mechanical strength of bone such that many subjects experience fractures while treated[4].

After the initial stages of GC-induced bone loss, a continued suppression of bone formation has been reported. This is mainly due to reduced activity of osteoblasts, the differentiated cells that are responsible for bone formation derived from a pool of SSCs. hPTH 1-34, also known as teriparatide, is FDA-approved to treat GC-induced osteoporosis (GIOP) as it stimulates osteoblasts to generate bone in the presence of GCs[5,6], decreases sclerostin production by osteocytes and prevents osteoclast maturation in the presence of GCs[7].

GC exposure can also induce osteonecrosis, the result of damaged skeletal vasculature that provides essential nutrients for normal cell function and tissue homeostasis. This adverse event is commonly associated with higher cumulative doses and longer treatment courses of systemic GCs; however, it has also been described after intra-articular injections, topical administration or low-dose, short-term oral GCs[3]. No major gender differences have been identified. While the exact etiology of osteonecrosis is not clear, a reduction in the vascular supply to the proximal femur is frequently observed. Angiogenesis is required for the osteogenic process, as endothelial cell invasion is needed to transition hypertrophic chondrocytes to osteoblasts at the growth plate, and angiogenesis through macrophage-initiated signaling is needed for the formation of a bone remodeling unit. However, the influence of GCs on angiogenesis is unclear[8].

Therefore, given the observation that GCs reduce bone formation and may alter vascular supply directly or indirectly to bone, the purpose of this study was to increase our understanding of how GCs affect the cellular and molecular components of the bone marrow environment. We performed a detailed analysis of the cellular composition, SSC activity and single cell gene expression in mice treated with GCs, after recovery from GCs and GC with hPTH 1-34 treatment. We determined that bone loss through GC exposure is initially associated with a reduced number of SSCs and vascular cells, while with continued GC exposure, skeletal precursor cells accumulated but lost their ability to differentiate into osteogenic and chondrogenic cell types. Interestingly, these SSCs secrete specific factors, including Basigin, that alter endothelial morphology and function through direct crosstalk. Concomitant hPTH 1-34 during GC exposure or genetic and antibody-mediated Basigin blockade abrogated that detrimental signaling axis and reversed the GC-mediated phenotype. Excitingly, anti-Basigin treatment also improved bone parameters in aged mice, suggesting a potential broader clinical utility. Altogether, these findings reveal a previously unappreciated connection between stem and endothelial cell interaction that can be targeted to prevent GC-induced bone loss.

## Results

### GC-induced bone loss is reversed by concurrent hPTH treatment

To assess the cellular and molecular changes of continuous GC exposure on bone tissue, we subcutaneously transplanted 4-month-old, male Balb/cJ mice with 5 mg Methylprednisolone (GC) or placebo release pellets. Randomly grouped mice were housed for 28 days before analysis. To assess the consequences of prolonged GC treatment, additional animals were kept for another 28 days and compared to two separate groups either having GC pellets removed after 28 days or animals receiving hPTH 1-34 at 40 µg/kg, 5 days/week during GC exposure throughout the experiment (Fig. 1a). As expected from our previous study using a similar model[9], micro-CT analyses and mechanical testing showed that 28- and 56-day GC exposure significantly reduced femoral trabecular bone parameters and bone strength, respectively, compared to placebo control (Fig. 1b–d, Supplementary Fig. 1a, b). Interestingly, removing GC pellets after 28 days did not reverse trabecular bone loss after an additional 28 days, while hPTH 1-34 in the presence of GC exposure normalized trabecular parameters to placebo levels. Cortical thickness and area were significantly reduced upon acute GC exposure but were unaltered in all groups monitored over a 56-day period (Supplementary Fig. 1c, d). To test how these observations were tied to changes in bone remodeling, we conducted dynamic histomorphometry and found that bone formation and osteoclast activity were inversely controlled by GCs. Mineral apposition and bone formation rate were significantly reduced by GC treatment at both investigated timepoints, and only hPTH 1-34 treatment improved those parameters to placebo control levels (Supplementary Fig. 2a–c). In contrast, osteoclast activity was strongly increased upon GC exposure as measured by TRAP staining and bone marrow-derived in vitro osteoclastogenesis; however, GC removal alone was sufficient to reduce bone resorption activity to control levels (Supplementary Fig. 2c, d). Osteoclast surface per bone surface was elevated in GC mice that received hPTH 1-34, suggesting stimulation of increased bone remodeling with high bone formation rates outweighing increased bone resorption. Altogether, these data provide insight into the complex effects of GC and hPTH 1-34 actions on bone parameters of our mouse model.

### GC exposure drives distinct cellular and molecular changes in bones

Having established our model of GC-induced bone loss, we next sought to derive a more detailed view of the cellular changes of bone tissue. We conducted 10X Chromium single-cell RNA-sequencing (scRNAseq) of dissected femurs from the four experimental groups on day 56. Given the low representation of non-hematopoietic cells in single cell preparations of bone tissue, we sorted equal numbers of CD45-positive and CD45-negative cell populations into the same collection tube for each group. Unbiased Leiden clustering analysis of stringently quality-filtered single cells established the cellular composition of captured cells for each experimental group (Fig. 1e, f). This approach covered the broad heterogeneous composition of the bone/bone marrow (BM) composition, including hematopoietic, mesenchymal and endothelial cell types. When we investigated specific differences in the cellular makeup of each group, we observed that GCs increased stromal cell populations in the BM, including CXCL12-expressing reticular (CAR) cells (Fig. 1g, h). There were also slight changes to committed bone-forming cell types, indicating potential alterations to mesenchymal lineage allocation commitment. Specifically, expression of genes associated with osteogenesis and chondrogenesis was reduced upon GC exposure but showed strong improvement to placebo levels if mice were simultaneously treated with hPTH 1-34 (Fig. 1i). Conducting global pathway enrichment analysis with the top 200 differentially expressed genes for each group, revealed that GC exposure led to increases in angiogenesis-related signaling in the BM environment (Fig. 1j). Specific gene expression patterns of endothelial genes showed that GC removal and hPTH 1-34 treatment normalized the expression to placebo levels (Fig. 1k). Similarly, while the immune cell compartment composition was not strongly altered by GC exposure, we observed an increase in pro-myeloid and pro-osteoclastic signaling that was partially reversed by GC removal and hPTH 1-34 treatment (Fig. 1l). In sum, scRNAseq of the bone tissue from the different experimental groups revealed

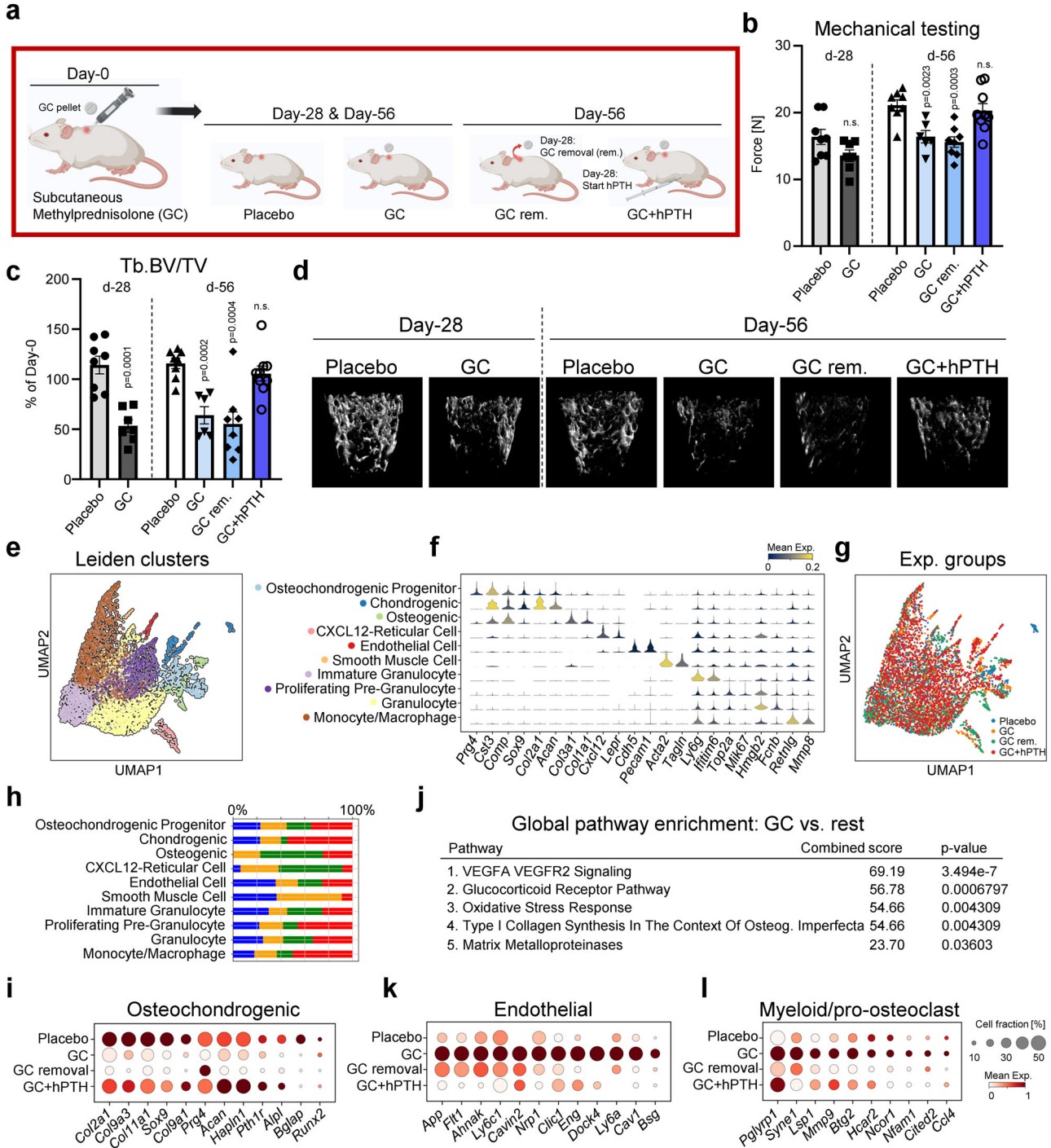

**Fig. 1 | A GC-mediated anti-osteogenic, pro-endothelial, pro-myeloid shift is reversed by hPTH. a** Experimental schematic for exposing mice to continuous glucocorticoids through subcutaneous Methylprednisolone treatment at two timepoints and varying interventions. Created in BioRender. Ambrosi, T. (2025) https://BioRender.com/8pmjxe4. **b** Mechanical strength test of femurs by 3-point bending. **c** Quantification of trabecular bone volume per total volume (Tb.BV/TV) shown as percentage change compared to day-0 age-matched controls. Analysis of femur bones from $n = 8$ biologically independent mice for Placebo and GC removal groups, $n = 7$ for GC d-28, $n = 6$ for GC d-56 and $n = 9$ for GC+hPTH. **d** Representative microCT images of distal femur trabecular bone at day-28 and day-56. Data shown as mean ± SEM. Statistical testing between Placebo and other group by two-sided unpaired Student's *t*-test. *$p < 0.05$, **$p < 0.01$ ***$p < 0.001$, ****$p < 0.0001$. **e** UMAP of femur-derived single cells from all experimental groups and their distinct clustering by Leiden. **f** Specific markers of Leiden clusters determining cellular identity. **g** UMAP plot showing cellular clustering labeled by experimental group. **h** Bar graphs showing the relative abundance of each cell type captured for each experimental group. **i** Dotplot showing selected osteochondrogenic gene expression in mesenchymal cell subsets. **j** Global pathway enrichment analysis of the top 200 differentially expressed genes in mesenchymal cells of the GC group using EnrichR. **k** Dotplots of endothelial and **l** myeloid gene expression in vascular and hematopoietic cell types, respectively, between different experimental groups.

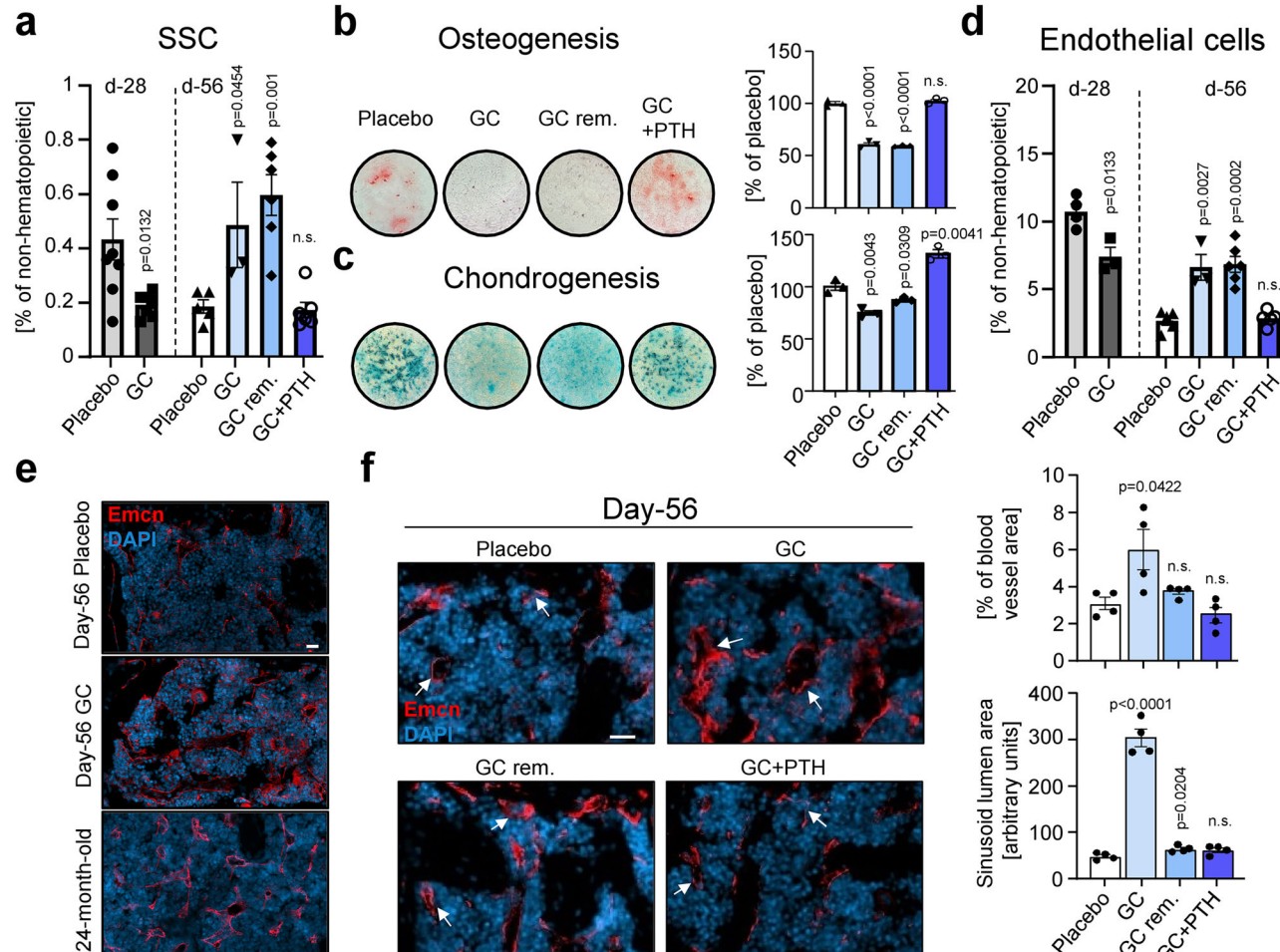

**Fig. 2 | GCs drive impaired skeletal stem and progenitor function and altered bone marrow blood vessel characteristics. a** Flow cytometry-based quantification of skeletal stem cell (SSC, CD45-Ter119-Tie2-CD90-6C3-CD105-CD51+[CD200+]) in femurs of experimental groups. $n = 8$ biologically independent mice for Placebo d-28 group, $n = 7$ for GC d-28 group, $n = 5$ for Placebo d-56 group, $n = 3$ for GC d-56 group, $n = 6$ for GC removal and GC+hPTH groups. **b** In vitro osteogenesis (Alizarin Red S stain) and **c** chondrogenesis (Alcian Blue stain) assays on purified, primary SSCs. Spectrophotometric quantification of staining (right). Cells from $n = 3$ biologically independent mice per group. **d** Flow cytometry-based quantification of endothelial cell populations (CD31+) in femurs of experimental groups. $n = 4$ biologically independent mice for Placebo d-28, $n = 3$ for GC d-28 and

d-56 groups, $n = 5$ for Placebo d-56 and GC+hPTH groups, $n = 6$ for GC removal group. **e** Immunohistochemistry staining for Endomucin (Emcn) in bone marrow of day-56 placebo and GC-treated mice, as well as in 24-month-old wild-type mice. **f** Representative immunohistochemistry staining for Endomucin (Emcn) in bone marrow of day-56 experimental groups and quantification of blood vessel area (right top) and size of sinusoid lumen (right bottom) based on immunohistochemistry staining for Endomucin. $n = 4$. Arrows: sinusoids. All data shown as mean ± SEM. Statistical testing between Placebo and other groups by two-sided unpaired Student's $t$-test. *$p < 0.05$, **$p < 0.01$ ***$p < 0.001$, ****$p < 0.0001$. Scale bars, 20 μm.

distinct cellular and molecular changes of the BM compartment supportive of alterations at the tissue level. Strikingly, the strongest changes were associated with alterations in angiogenic signaling.

## GCs alter skeletal stem and progenitor function and blood vessel characteristics

Since we observed bone loss and reduced expression of osteochondrogenic genes in mice exposed to GC, we wondered if skeletal stem and progenitor activity might be altered. Therefore, we analyzed the frequency of phenotypic SSCs (CD45-Ter119-Tie2-CD90-6C3-CD105-CD51+[CD200+]) and the directly downstream transient multipotent bone-cartilage-stromal progenitors (BCSPs; CD45-Ter119-Tie2-CD90-6C3-CD105+CD51+) using flow cytometry[10]. Results showed that GCs initially (day 28) decreased SSC and BCSP numbers, while extended exposure (day 56) drove an increase in SSCs (Fig. 2a, Supplementary Fig. 3a, b). Similarly, GC removal did not change the significant accumulation of SSCs and BCSP compared to placebo controls. Again, only GC with concomitant hPTH 1-34 normalized skeletal progenitor

abundance to control levels. To further assess the functional properties of SSCs, we freshly purified them from the bone tissue of each experimental group on day 56 and seeded them for standard in vitro osteogenic and chondrogenic assays. In alignment with micro-CT and transcriptomic data, GCs significantly reduced the osteogenic and chondrogenic potential of skeletal progenitors compared to placebo controls, which was also not altered upon GC removal assessed after 28 days (Fig. 2b, c). SSCs from bones of mice exposed to GC but treated with hPTH 1-34 injections showed strong osteochondrogenic activity. These results indicate that skeletal stem and progenitor function is negatively affected by GCs, directly associating it with negative structural and mechanical changes observed for bones.

Based on scRNAseq results, we also wondered how endothelial numbers and composition may be altered. Flow cytometric analysis revealed a similar pattern as seen with SSCs. While after 28 days, endothelial cell numbers were reduced with GCs compared to the placebo group, longer GC exposure drove an accumulation of endothelial cell presence that was only normalized to placebo levels when mice received hPTH 1-34 with GCs (Fig. 2d, Supplementary Fig. 3c). When we

stained histological sections of femur bones after 56 days of GC exposure with the endothelial marker Endomucin, we observed an increase of blood vessel abundance as well as distinct morphological changes that resembled the endothelial compartment of aged (24-month-old) mice (Fig. 2e). Previous work has reported that cellular stress in the BM environment is connected to increased numbers of sinusoids with dilated lumina[11]. Quantitative analyses showed that at day 56 of GC exposure, there were significantly more Endomucin-positive sinusoidal blood vessels with increased lumen area in the BM compared to placebo controls (Fig. 2f, Supplementary Fig. 3d). The blood vessel area and dilation of lumens remained at placebo levels when mice were concomitantly treated with hPTH 1-34. In summary, GCs mediate specific changes to bone-forming lineage cells and angiogenic processes that can be reversed by co-stimulation with hPTH 1-34 treatment, suggesting a direct connection between SSCs and endothelial cells.

## GCs alter SSC-mediated ossicle formation

To determine if GC-mediated alterations were connected through changes observed in SSC and endothelial cell activity, we assessed SSC-specific bone formation characteristics via renal capsule transplantation[12]. To that end, we freshly FACS-purified equal numbers of SSCs from ubiquitous GFP-reporter mice and transplanted them beneath the renal capsule of wild-type C57BL/6 mice. We randomly assigned mice to four groups, i.e., a control group receiving placebo pellets at the time of surgery, a group with hPTH 1-34 treatments but no GCs as well as two groups receiving GC release pellets. One of the GC groups received 5x weekly hPTH 1-34 over the 21 days of the experiment (Fig. 3a). At that time, we isolated tissue grafts and processed them for scRNAseq. As expected, generated ossicles of all groups contained mesenchymal, endothelial and hematopoietic cell types recognized by distinct single cell clustering based on overall gene expression profiles (Fig. 3b–e). Given limitations of scRNAseq approaches, including transcriptomic coverage, preventing faithful separation of rare, homogeneous SSC populations using unbiased clustering analysis, we identified the SSC-enriched cluster by known but not unique marker expression (e.g., *Sox9*, *Col2a1*, *Pthlh*) within a broader pool of skeletal stem and progenitor cells (SSPCs). These SSPCs also showed high expression of the gene Nr3c1 encoding the glucocorticoid receptor compared to other mesenchymal lineage cells (Supplementary Fig. 3e). Mice treated with GCs alone presented with a higher number of undifferentiated SSPCs than placebo-treated mice. The expression of osteogenic and chondrogenic gene programs also decreased with GC treatment alone but was rescued by hPTH 1-34 treatment (Fig. 3f). Analyzing gene expression patterns of all cells revealed that GC exposure led to high expression of genes related to cellular stress including those associated with the senescence-associated secretory phenotype (SASP) and oxidative stress, supporting the idea that GCs appear to drive an aging-like and pro-inflammatory microenvironment (Fig. 3g). Interestingly, genes involved with blood vessel recruitment and formation were highly increased in the presence of GCs which also was associated with increased oxidative stress gene expression (Fig. 3h). Once again, hPTH 1-34 added to the GC treatment was able to reverse gene expression patterns toward placebo control levels. Of note, higher circulating levels of hPTH 1-34 elevated expression of SASP/cellular stress-related genes compared to the placebo group. Slight increases in those markers might be the reflection of a more activated state of hPTH 1-34 exposed cells rather than increased inflammation. Since SSC-derived ossicle formation strongly depends on host endothelial blood vessel recruitment, these results support a co-regulation of SSC and endothelial lineages during GC exposures and provide a list of molecular interaction partners.

## GC-induced SSC-derived Basigin impairs bone-forming and vascular network properties

Among the multiple pro-endothelial genes upregulated upon GC exposure and rescued by hPTH 1-34 treatment (*Kdr/Vegfr*, *Vegfa*, *Pdgfa*,

*Wnk1*), we identified *Basigin* (*Bsg*) as well as its known intracellular interaction partners of the monocarboxylate transporter (*Mct1-3*) family to be highly expressed in accumulated SSPCs (Figs. 1k, 3h, 4a, Supplementary Fig. 3f)[13]. We therefore hypothesized that Basigin, a known activator of cell proliferation[14,15], might drive GC-mediated aberrations in the BM environment through a distinct SSC-endothelial signaling axis. To further explore this concept and its translational relevance, we conducted functional tests in primary human SSCs (hSSCs, CD45⁻CD235a⁻CD31⁻TIE2⁻CD146⁻PDPN⁺CD164⁺CD73⁺) and in the human VeraVec HUVEC endothelial cell line. We collected 48 h supernatant from hSSCs either lentivirally overexpressing Basigin or vector-controlled to directly tie SSPC-derived Basigin expression to changes in endothelial cell activity. Then, we assessed the effect of the supernatants of both groups during in vitro tube formation and wound scratch assays of VeraVec endothelial cells. Strikingly, in the presence of high levels of Basigin in the supernatants, blood vessel architecture was significantly impaired, as indicated by reduced mesh, tube and node numbers compared to the control (Fig. 4b, c). We also found that Basigin impaired endothelial cell migration in a wound scratch assay (Fig. 4d). In addition, elevated Basigin levels increased ROS generation in cultured endothelial cells (Fig. 4e), confirming scRNAseq readouts (Fig. 3g, h). Furthermore, overexpression of Basigin in hSSCs drove increased colony-forming ability, which is a measure of proliferative activity, while it impaired in vitro osteogenic and chondrogenic differentiation compared to controls (Fig. 4f–h). Finally, we subcutaneously transplanted equal numbers of hSSCs either overexpressing Basigin or vector control into immunodeficient NOD scid gamma (NSG) mice to determine if ossicles formed displayed differences in osteogenesis and angiogenesis in vivo (Fig. 4i, j). In line with a recent report on the effect of GCs on fracture healing[16], we observed reduced bone remodeling dynamics in grafts derived from Basigin-overexpressing cells that were less mineralized and showed low bone resorption activity by host-derived osteoclasts (Fig. 4k, l). Interestingly, the shortened morphology of recruited blood vessels of the Basigin-SSC grafts mirrored the GC-induced phenotype observed in femurs (Fig. 4m). These results establish a direct connection between SSC-derived Basigin and detrimental effects of skeletogenesis and blood vessel architecture caused by GCs that are transferable to human cells.

## Antibody-mediated blockade of hSSC-derived Basigin reverses impaired endothelial function and bone formation in vitro

Next, we investigated whether the negative effects of Basigin overexpression in human SSCs could be pharmacologically reversed. To that end, we treated human SSCs with a monoclonal antibody against Basigin (aBSG) and collected supernatant to test its effect on vascular modeling. Indeed, the detrimental paracrine effect on human endothelial tube formation in the presence of supernatant from Basigin-overexpressing SSCs was mostly returned to control levels with antibody treatment (Fig. 5a). Similarly, VeraVec cells exposed to Basigin and treated with aBSG performed similarly to controls in wound scratch assays (Fig. 5b). Strikingly, the Basigin induced osteogenesis-impairing effects due to overexpression in SSCs were rescued when we either exposed the cells to hPTH 1-34 or aBSG (Fig. 5c), supporting Basigin antibody blockade as a strategy to prevent or reverse GC-induced SSC dysfunction mediated bone loss.

## Pharmacological and genetic ablation of Basigin in vivo prevents detrimental bone loss induced by GCs

To confirm our in vitro findings, we next tested whether antibody blockade of Basigin can prevent GC-induced bone loss in mice. We compared placebo-treated control mice with mice that were exposed to GCs for 28 days. Of these GC-treated mice we looked at mice receiving no additional therapy or treated with hPTH 1-34 or aBSG throughout that time period (Fig. 6a). Histological analyses of femoral

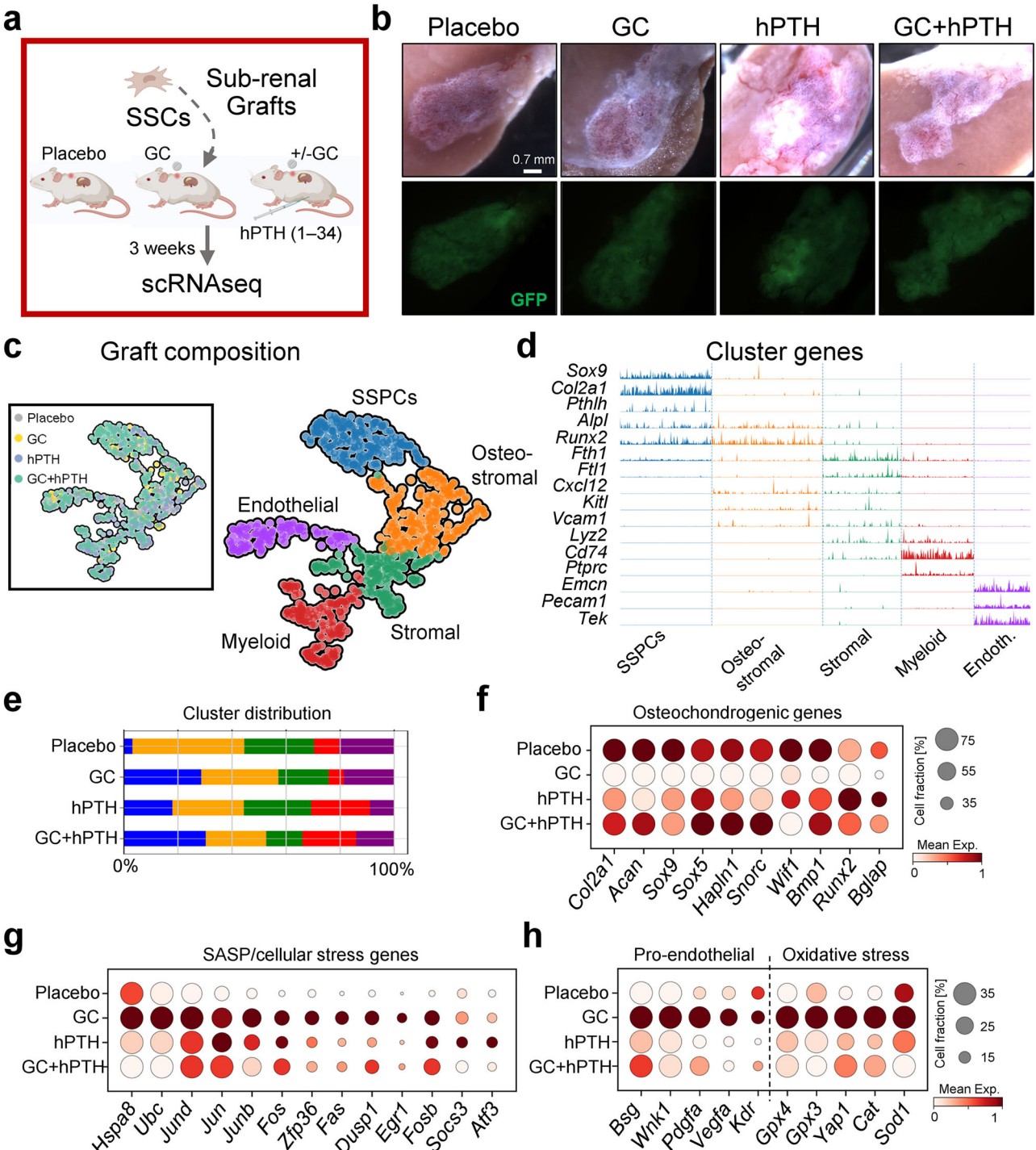

**Fig. 3 | GC alters de novo in vivo bone formation by SSCs and the niches they generate. a** Experimental schematic of renal capsule transplants of freshly purified wild-type, GFP-labeled SSCs in mice exposed to placebo, GC, hPTH alone, or GC +hhPTH (hPTH 1-34). Created in BioRender. Ambrosi, T. (2025) https://BioRender. com/r8dp0jv. **b** Light microscopic images (top) and GFP signal of grafts formed beneath the renal capsule of transplanted mice. **c** Single-cell transcriptomic analysis of formed tissue grafts displayed as UMAP of single cells clustered by Leiden for cell type separation and by group (top left). SSPCs, skeletal stem and progenitor cells. **d** Trackplot of specific markers of Leiden clusters determining cellular identity. **e** Bar graphs showing the relative abundance of each cell type captured for each experimental group. **f** Dotplot showing selected osteochondrogenic (left) genes in SSPC cluster between different experimental groups. **g** Dotplot showing SASP (senescence-associated secretory phenotype)/cellular stress-related gene expression in all cells between different experimental groups. **h** Dotplot showing pro-endothelial and oxidative stress-related gene expression between different experimental groups.

bone sections at day 28 showed that hPTH 1-34 and aBSG treatments reduced Basigin expression seen in the GC only group (Fig. 6b). Also, the reduction in Basigin levels in these treatment groups correlated with trabecular bone volume, osteoclast numbers and bone marrow endothelial morphology of placebo controls (Fig. 6c–e), prevented the

GC-induced short-term reduction in SSC frequency (Fig. 2a) and maintained their in vitro osteogenic potential (Fig. 6f, g). While we observed a GC-driven shift toward an increase in circulating myeloid cell types in blood, which was reversed by aBSG treatment, we did not detect any significant differences in the hematopoietic stem and

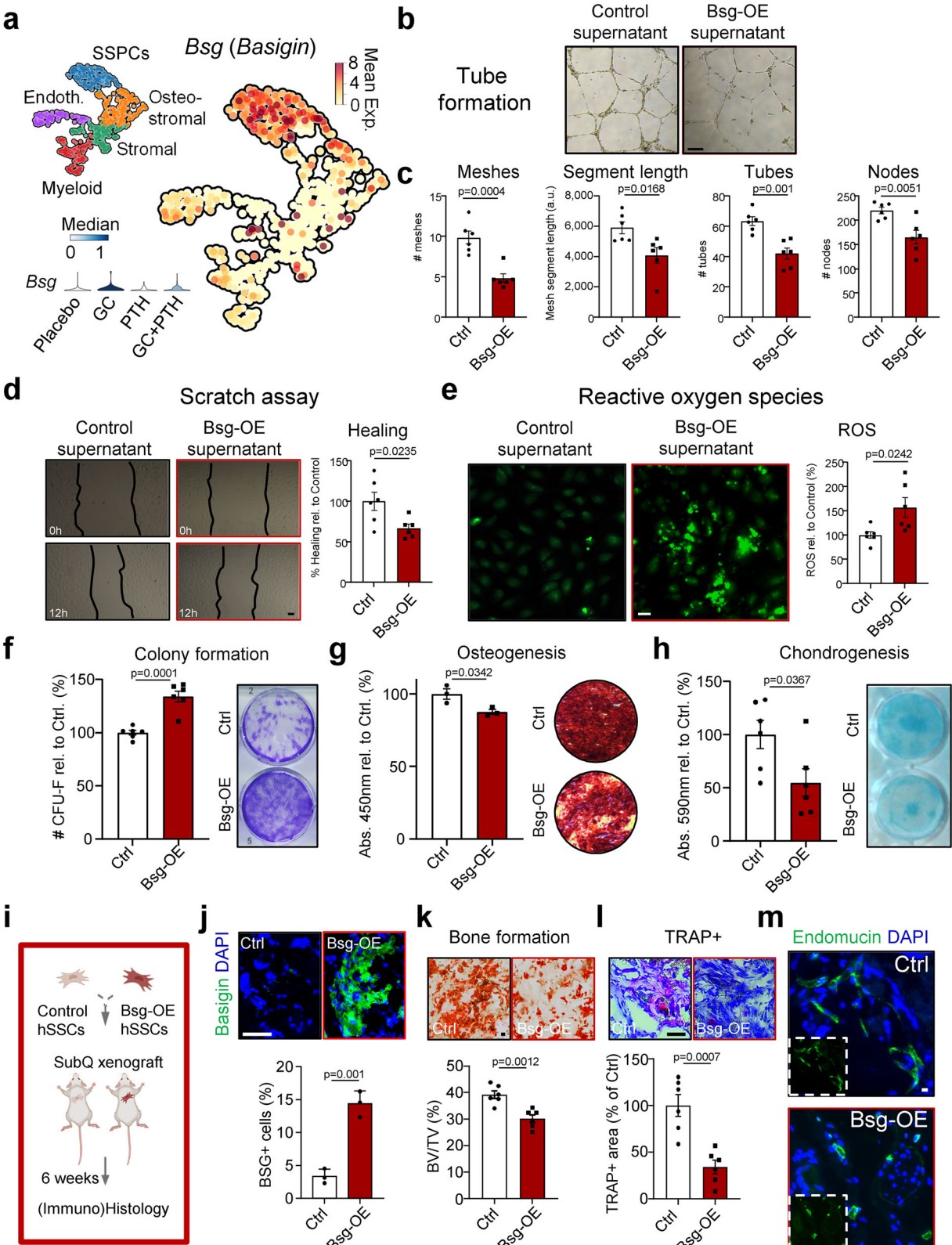

progenitor compartment in the bone marrow (Fig. 6h–i, Supplementary Fig. 4).

To strengthen our findings and tie GC-induced bone loss specifically to skeletal stem and progenitor cell-derived Basigin, we crossed Collagen 2a1 (Col2a1)-Cre^ERT2 with Basigin^flox/flox (Bsg^fl/fl) mice to generate conditional knockout mice lacking Basigin expression in SSPCs. Our previous studies[17,18] and current results (Fig. 3d)

indicated that SSPCs, but not endothelial or immune cell types, express *Col2a1*. Female Col2a1-Cre^ERT2 x Bsg^fl/+ transgenic mice were tamoxifen-induced when they reached 8 weeks of age and implanted with 4-week GC release pellets 1 week later (Fig. 6j). Compared to vehicle treated (corn oil) control mice, heterozygous SSPC-specific knockout mice presented with significantly reduced Basigin levels upon GC exposure (Fig. 6k). This reduction was accompanied by

**Fig. 4 | Basigin overexpression alters human skeletal lineage dynamics that impair endothelial function. a** Single cell transcriptomic analysis of tissue grafts from mouse SSCs either exposed to placebo, GC, human PTH or GC+human PTH (hPTH 1-34) displayed as UMAP showing high expression of Basigin in SSPCs. SSPCs, skeletal stem and progenitor cells. Left bottom: Violin plot showing increased expression of *Bsg* in GC group across all cells. **b** Representative image of tube formation assay by VeraVec endothelial cells exposed to control supernatant or supernatant of human SSCs overexpressing Basigin. **c** Quantification of ImageJ-based tube formation analysis. *n* = 6 independent samples from two independent experiments. **d** Representative brightfield images of the endothelial scratch assay (left) and its quantification (right). *n* = 6 independent samples from three independent experiments. **e** Measurement of reactive oxygen species in cultured endothelial cells after 24 h supernatant exposure. Left: representative fluorescence images. Right: Quantification of fluorescence signal. *n* = 6. All experiments with supernatant from at least two donors and two independent experiments. **f** Colony-forming unit ability of primary human SSCs either overexpressing control vector or Basigin. CFU-Fs stained by Crystal violet. *n* = 6 independent replicates from cells of two biologically independent donors. **g** In vitro osteogenesis (Alizarin Red S stain)

by hSSCs of the same groups. *n* = 3 independent replicates from cells of one biological donor. **h** In vitro chondrogenesis (Alcian Blue stain) assays of the same groups. *n* = 6 independent replicates from cells of two biologically independent donors. **i** Schematic of subcutaneous transplant approach. Created in BioRender. Ambrosi, T. (2025) https://BioRender.com/4wl26h3. **j** Immunohistochemistry of Basigin (green) expression in SSC-generated grafts with corresponding quantification. *n* = 3 biologically independent donor cells in three biologically independent mice per group. **k** Representative Alizarin Red S staining of sectioned grafts and quantification of mineralized tissue in grafts containing transplanted human SSCs as assessed by Alizarin Red S staining. *n* = 6 sections from three biologically independent grafts. **l** Representative TRAP staining of sectioned grafts and quantification of TRAP-positive area of grafts. *n* = 6 sections from three biologically independent grafts. **m** Representative immunohistochemistry staining for Endomucin and DAPI of sectioned graft. Small insert shows Endomucin staining without DAPI. All data shown as mean ± SEM. Statistical testing between Placebo and other groups by two-sided unpaired Student's *t*-test. *$p < 0.05$, **$p < 0.01$ ***$p < 0.001$, ****$p < 0.0001$. Scale bars, 50 μm.

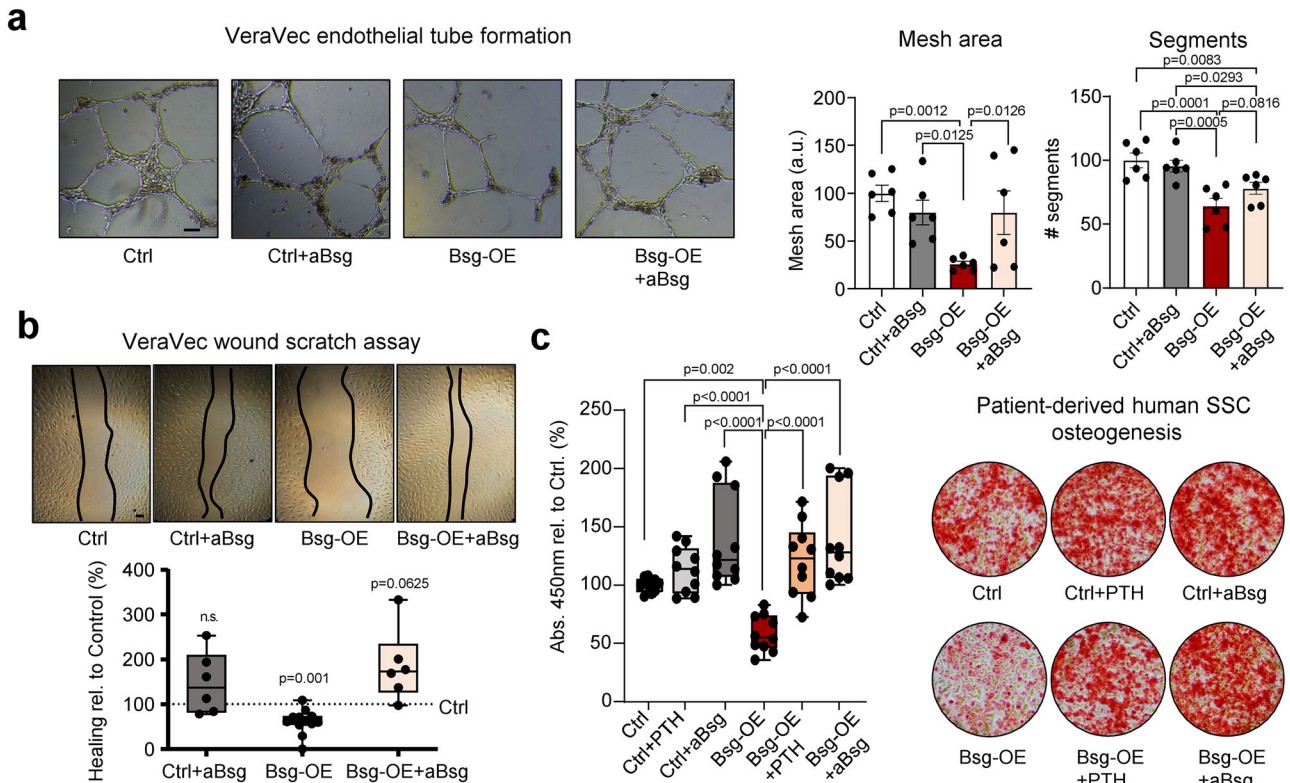

**Fig. 5 | Antibody blockade of Basigin rescues GC-induced endothelial and skeletal impairments in vitro. a** Tube formation assay of human VeraVec endothelial cells treated with supernatant from different experimental groups. *n* = 6 independent replicates per group from two independent experiments. **b** Representative brightfield images of endothelial scratch assay (top) and its quantification displayed as boxplots (bottom) after 12 h. Endothelial cells were treated with supernatant of cultured hSSCs treated as shown. *n* = 6 independent replicates from two independent experiments for Ctrl+aBsg and Bsg-OE+aBsg groups. *n* = 12 independent replicates from three independent experiments for

Bsg-OE group. Boxplots with box and whiskers Min to Max. **c** In vitro osteogenesis of patient-derived human SSCs. Left: Boxplots showing quantification of Alizarin Red S staining. Right: Representative images of Alizarin Red S staining. Ctrl: control media only; Ctrl+PTH: control media with PTH 1-34 treatment 6 h before media change; Ctrl+aBsg: control media with Basigin antibody treatment. Bsg-OE: Lentivirally Basigin-overexpressing human SSCs with control media. *n* = 10 independent replicates of cells from two independent donors. Data shown as mean ± SEM. Statistical testing in a,c by one-way ANOVA with Fisher-LSD test, in b by Wilcoxon signed-rank test. *$p < 0.05$, **$p < 0.01$ ***$p < 0.001$, ****$p < 0.0001$. Scale bars, 100 μm.

significantly higher trabecular bone parameters and reduced osteoclast numbers suggesting that a reduction in expression of Basigin from SSPCs was sufficient to prevent GC-induced bone loss (Fig. 6l, m, Supplementary Fig. 4e–g). Thus, administration of aBSG during GC treatment or conditional genetic ablation specifically in the skeletal stem and progenitor lineage prior to GC exposure prevents skeletal maladaptation.

### Basigin antibody therapy improves bone mass in aged mice independent of sex

Since Basigin has been reported to be a therapeutic target for a number of pro-inflammatory and pro-fibrotic conditions, we asked whether aBSG could also reverse age-related bone loss, i.e., osteoporosis[19,20]. Indeed, immunohistochemistry analysis showed higher expression of Basigin in bone marrow of old mice, in particular

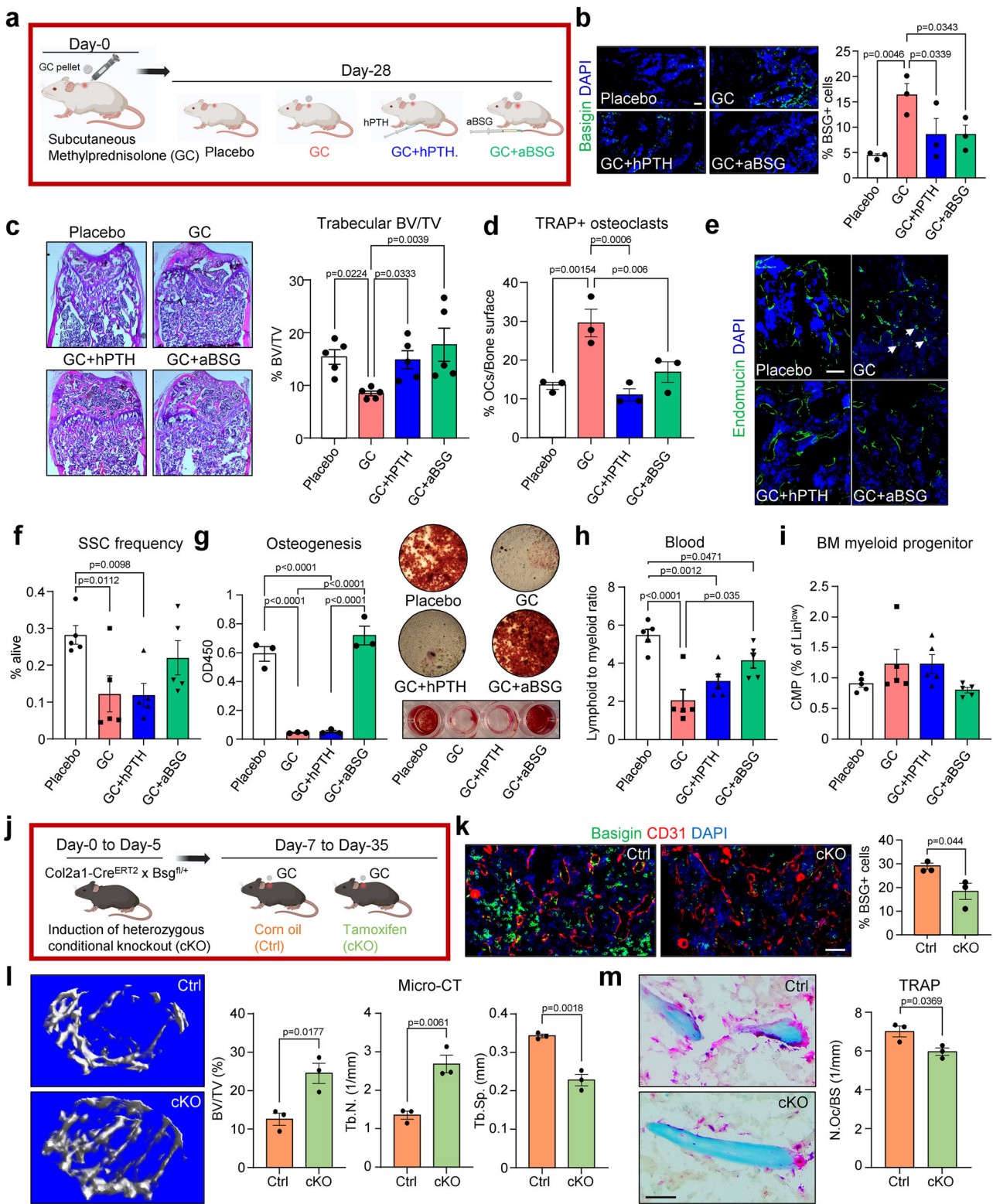

in females (Fig. 7a). When we treated 2-year-old female and male mice with aBSG three times/week at 1 mg/kg for 4 weeks we found a significant improvement in volumetric trabecular bone of long bones compared to IgG control treated mice (Fig. 7b). Vertebral L5 bone mineral density at 14 days and BV/TV at 28 days of treatment measured by DEXA and micro-CT, respectively, demonstrated an anabolic effect on the skeleton by aBSG independent of sex (Fig. 7c, d). Interestingly, SSCs of the same mice showed higher in vitro osteogenic potential,

whereas histological analysis suggested an increase in bone resorptive activity with aBSG (Fig. 7e, f). This was accompanied by endothelial restoration but no detectable changes in the frequency of skeletal and hematopoietic stem and progenitor cells (Supplementary Fig. 5). Collectively, these results indicated that aBSG treatment boosts bone remodeling in aged mice in a sex-independent manner, leading to a net gain in bone mass despite elevated osteoclast activity likely through higher bone-forming activity.

**Fig. 6 | Pharmacological blockade and genetic knockout of Basigin reverses GC-induced skeletal changes. a** Schematic of experimental setup. Created with BioRender.com. **b** Representative immunohistochemistry images and quantification of Basigin expression in the different experimental groups. $n = 3$ biologically independent mice per group. **c** Representative H&E stain of distal femurs of experimental groups with quantification of trabecular bone area below the growth plate region. $n = 5$ biologically independent mice per group. **d** TRAP staining and quantification of femur bones from all experimental groups. $n = 3$ biologically independent mice per group. **e** Representative images of Endomucin-positive endothelium in the bone marrow environment. Arrows: small sinusoids. **f** Flow cytometric analysis of femur bones of experimental groups for skeletal stem cells (SSCs). $n = 5$ biologically independent mice per group. **g** In vitro osteogenic differentiation of bone marrow stromal cells derived from experimental groups stained with Alizarin Red. $n = 3$ biologically independent mice per group. **h** Flow cytometric analysis of blood from mice of different experimental groups showing lymphoid to myeloid ratio. **i** Flow cytometric analysis of bone marrow (BM) common myeloid progenitor cells (CMPs) from mice of different experimental groups. $n = 5$ biologically independent mice per group. **j** Experimental schematic for GC experiments in heterozygous Basigin knockout mice. Control (Ctrl) and tamoxifen-treated conditional knockout (cKO) mice were compared for their skeletal response to GC exposure. Created in BioRender. Ambrosi, T. (2025) https://BioRender.com/35tbvy1. **k** Co-staining of Basigin (green) and endothelial marker CD31 (red) in bones of experimental groups after GC exposure. **l** Micro-CT analysis of trabecular bone parameters at endpoint. $n = 3$ mice per group. **m** TRAP staining for osteoclast activity at endpoint. Average from $n = 3$ biologically independent mice per group. All data shown as mean ± SEM. Statistical testing by one-way ANOVA with Fisher-LSD test for pharmacological approach (**b–i**) or by two-sided unpaired Student's $t$-test for comparison of genetic knockout groups (**k–m**). $*p < 0.05$, $**p < 0.01$, $***p < 0.001$, $****p < 0.0001$. Scale bars, 50 μm.

## Discussion

In this study, we identified a previously unknown connection between SSC function and angiogenesis in GIOP. While it has been established that GCs increase osteoclastogenesis and alter osteogenesis and angiogenesis, a mechanism related to vascular changes has not been appreciated. Our findings demonstrate that continuous treatment with GCs promotes SSC proliferation, leading to their accumulation at the expense of differentiation required for bone formation. We uncovered a signaling axis by which Basigin, a transmembrane protein, specifically expressed by skeletal stem and progenitor cells when exposed to GCs, alters endothelial function and remodels BM vasculature, driving cellular stress and promoting bone loss through reduced stem cell-based bone formation and increased resorption.

Basigin has been considered an innovative marker in the context of cancer stem cells[21], but little is known about Basigin in the realm of skeletal biology. Two studies report that high levels of Basigin promote osteoclastogenesis through NFATc1 signaling[22] and its involvement in alveolar bone remodeling and soft tissue degradation[23]. It also acts as a potent stimulator of Interleukin-6 secretion in multiple cell lines that include monocytes[24], altogether supporting our observation of increased pro-osteoclastic signaling and bone resorption activity in GC-treated animals with high stem and progenitor cell-specific Basigin expression. In line with SSC accumulation as well as their altered cytokine and elevated extracellular matrix factor expression upon GC exposure, Basigin has also been found to contribute to tumor progression by stimulation of proliferation, elevated growth factor secretion and its matrix metalloprotease activity[25]. More specifically, previous studies have shown that Basigin acts as a chaperone for the shuttling of monocarboxylate transporters (MCTs) to the cell surface, where MCTs facilitate the movement of metabolites such as short-chain fatty acids and lactate. Lactate can directly regulate the cell cycle to promote cellular proliferation[26], for example, by modifying histones through lactylation changing epigenetic states[27]. Indeed, in Basigin-expressing SSPCs, we found concomitant upregulation of *Mct1-3* needed for lactate influx but downregulation of *Mct4* used for lactate efflux, suggesting that the altered metabolic state of SSPCs with high intracellular lactate could contribute to the functional changes observed. Follow-up studies will have to experimentally address this and potential other mechanisms, such as altered mechano-sensing of SSPCs due to Basigin-mediated ECM remodeling[28].

At the endothelial cell level, we found that GC-induced vascular impairments are closely tied to SSC-derived Basigin. Supporting our findings connecting Basigin to blood vessel formation is its known role as a coreceptor for vascular endothelial growth factor receptor 2 (Kdr/Vegfr2) in endothelial cells, enhancing their Vegfr-mediated activation and downstream signaling[29]. Continuous GC treatment in mice increased endothelial cell number and blood vessel area. However, Basigin exposure to endothelial cells resulted in irregularities in the connectivity and thickness of the nascent blood vessels. Flow cytometric and histological readouts, single-cell gene expression data and functional experiments connect these vascular changes to enhanced Vegfr signaling with a pathological blood vessel phenotype and increased oxidative stress. This is corroborated by studies that have shown that overactivation of Vegfr signaling is a driver of cellular stress in part through the stimulation of cell-damaging and inflammatory reactive oxygen species that negatively affect formation, migration and permeability of blood vessels[30,31]. One of the most perplexing questions in medicine is how glucocorticoids induce osteonecrosis (ON). While epidemiologic studies find GCs increase the risk of ON, the pathology of the ON lesion is that of reduced blood vessel density and increased adipose tissue within the bone marrow[8]. Therefore, our observations may be relevant to GC-induced ON, as the altered morphology of these nascent blood vessels may reduce the ability to deliver hemoglobin/oxygen, and likely other crucial nutrients, to the bones. In certain areas of specific skeletal compartments, like the femoral head, this may lead to increased cell death. Potentially connected to this, low hemoglobin is independently associated with increased fracture risk in non-vertebral bones of older adults[32]. Altogether, it will be worthwhile to investigate if the pronounced changes in angiogenic signaling mediated by Basigin may initiate the development of GC-induced ON and higher fracture risk.

Excitingly, the observation that this pathological signaling mechanism is reversible presents therapeutic vantage points to counter GC-induced bone loss and maybe ON. Concomitant administration of hPTH or aBSG prevented bone loss and other pathological skeletal phenotypes observed with GC exposure. Counterintuitively to the known anti-inflammatory actions of GCs, one of the bone marrow-specific changes we observed with GC exposure was an increase in myeloid signaling. This result is consistent with the bone cell activity changes observed in GC-treated subjects, as GC treatment reduces bone formation and increases the number and activity of osteoclasts[8,9]. The locally bone-restricted pro-inflammatory marker upregulation, often also seen with cellular stress and SASP, is likely indirect and related to changes in bone formation dynamics that favor increased osteoclastogenesis and pathological remodeling of blood vessel architecture that drive disruption in oxygen supply, favoring pro-necrotic states at least in part through Basigin-mediated mechanisms. It remains to be determined if strategies preventing GC-induced bone loss have any impact on the intended clinical effects of GC therapy.

We observed that the withdrawal of GCs on day 28 did not restore bone mass to baseline levels by day 56. Elevated GCs result in a reduction in the release of GCs from the hypothalamus[33], and our 28-day GC treatment may have sufficiently suppressed the hypothalamic-pituitary-adrenal (HPA) axis such that the 28-day treatment period was insufficient to see recovery of the skeletal tissues. In support of this observation, both osteogenesis and angiogenesis measurements for this group were similar to those of the GC-only group at 56 days. New studies will need to investigate the HPA axis recovery in this treatment

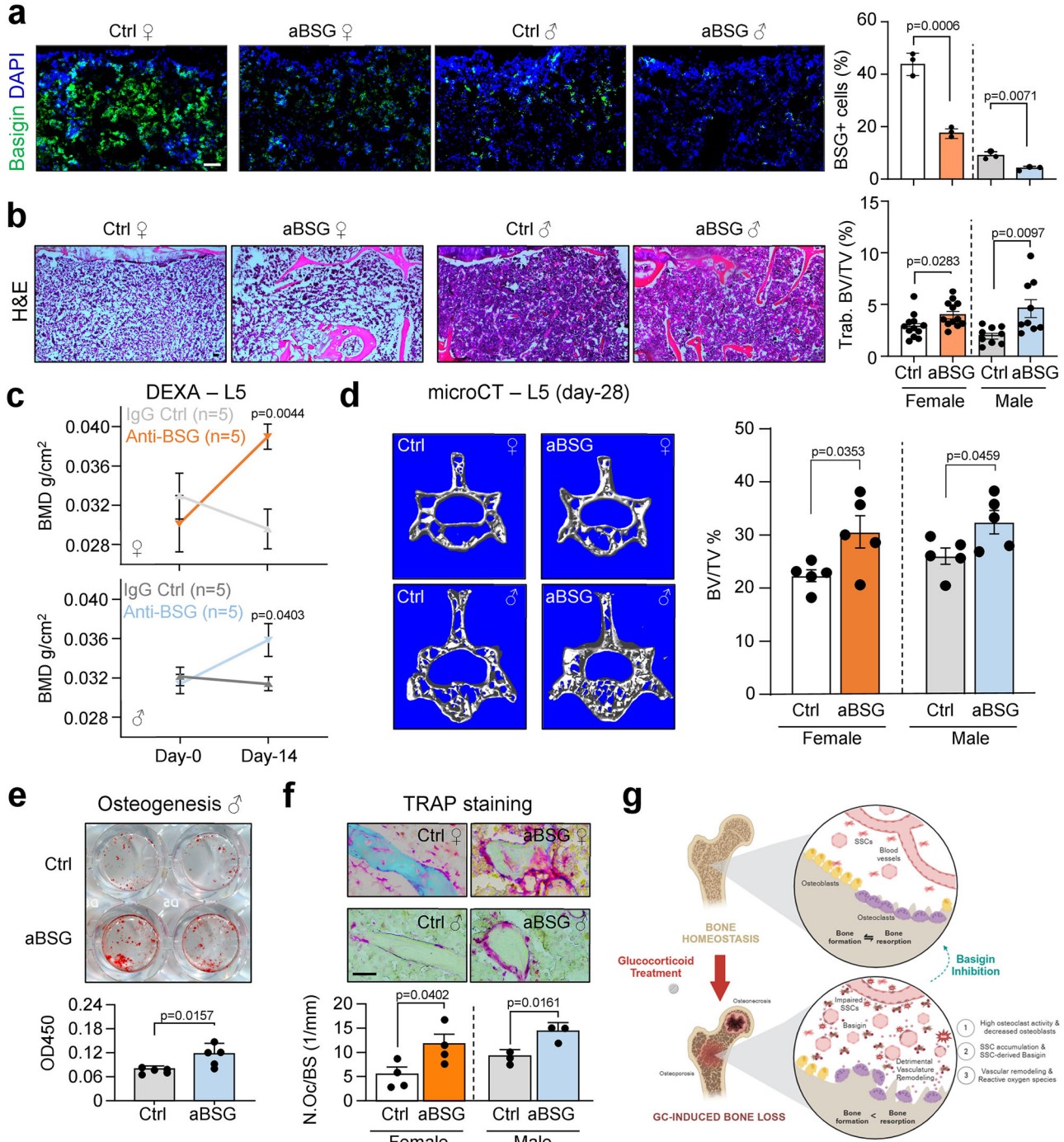

**Fig. 7 | Antibody blockade of Basigin in aging mice reinvigorates skeletal remodeling improving bone mass. a** IHC of Basigin expression in tibias of 24-month-old mice. Right: Histology-based quantification Basigin-expression cells per analyzed area. $n = 3$ biologically independent mice per sex and group. **b** H&E staining of proximal tibia regions below the growth plate. Right: Histology-based quantification of tibial trabecular BV/TV. $n = 12$ biologically independent female mice per group. $n = 9$ biologically independent male mice per group. **c** DEXA bone mineral density (BMD) measurements of vertebral L5 at treatment starting day-0 and at day-14 of treatment. **d** Micro-CT measurement of vertebral L5 BV/TV at 4 weeks of treatment. $n = 5$ biologically independent mice per sex and group.

**e** Representative Alizarin Red S staining of in vitro osteogenesis. Cells from $n = 5$ independent mice per group. **f** TRAP quantification in tibia sections of experimental groups. N.Oc/BS: number of osteoclasts per bone surface. Analysis of $n = 4$ biologically independent female mice per group. $n = 3$ biologically independent male mice per group. **g** Graphical summary of experimental findings. Created in BioRender. Ambrosi, T. (2025) https://BioRender.com/03daips. All data shown as mean ± SEM. Statistical testing between IgG control and other group by two-sided unpaired Student's $t$-test. $*p < 0.05$, $**p < 0.01$, $***p < 0.001$, $****p < 0.0001$. Scale bars, 50 μm.

group to contextualize our current findings. Clinical studies have reported that GIOP can be treated with hPTH 1-34, leading to an increase in trabecular bone mass and biochemical markers of increased bone formation. Furthermore, meta-analyses of multiple

studies have reported improved bone strength[6,34]. However, the effect of hPTH 1-34 on angiogenesis in the presence of GCs has not been investigated. Our study demonstrated that hPTH 1-34 treatment during GC exposure reduced Basigin expression, restored blood vessel

number and morphology, as well as osteochondrogenic differentiation of SSCs. The role of hPTH 1-34 in angiogenesis is complex and might be further affected by GC exposure. While hPTH 1-34 has previously been shown to be osteoanabolic and to direct blood vessels toward the bone-forming surface, it has not been found to be pro-angiogenic[35]. In contrast, direct beneficial actions of hPTH 1-34 on endothelial cells have been reported in fracture regeneration settings[36,37]. Even though GCs and hPTH 1-34 likely act on multiple skeletal lineage cell populations, our results provide evidence for their crucial role in regulating SSC activity.

Intriguingly, we could show that the antibody treatment against Basigin and heterozygous conditional deletion of SSPC-derived Basigin mitigate both its detrimental autocrine and paracrine effects on skeletogenesis and vascular modeling, respectively. Recognizing that many phenotypic changes in response to GC treatment resembled age-associated skeletal changes, i.e., dysfunctional SSCs, pro-osteoclastic signaling and loss of bone marrow blood vessel integrity, we deduced that GC-induced bone loss might underlie overlapping mechanistic processes[17,38]. Fittingly, we found Basigin to be highly abundant in the bone marrow of 2-year-old mice. When we tested treatment with aBSG in aged male and female mice, bone parameters significantly improved, suggesting antibody treatment against Basigin could also serve as an anti-osteoporotic drug independent of sex. Additional work with variations in dosing and longer treatment regimens is needed to validate this. Similar to hPTH 1-34, aBsg might restore proper SSC function, thereby preventing elevated Basigin levels and their negative downstream effects. Future work will have to inquire if aBSG elicits overlapping molecular pathways in SSCs or if it might be a superior therapeutic approach to hPTH 1-34 to prevent GC-mediated bone loss. Given that GC treatment is often prescribed to patients for many years and hPTH 1-34 can only be administered for a few years to restore bone mass, aBSG might become a valuable alternative. In summary, our work is the first to report the detrimental role of Basigin signaling in osteo-angiogenic coupling of bones and provides a therapeutic target to prevent or potentially reverse skeletal health decline. Future studies are needed to determine the specific autocrine and paracrine signaling cascades Basigin elicits on skeletal and endothelial cell populations in more detail, and whether its blockade might also be able to reverse established GIOP and prevent GIOP-induced ON.

## Methods
### Animals
All mouse experiments complied with all relevant ethical regulations and were conducted under approved protocols by the Institutional Animal Care and Use Committee of Stanford University and the University of California, Davis. Mice were maintained at animal facilities in accordance with institutional guidelines. Mice were given food and water ad libitum and housed in temperature-, moisture- and light-controlled (12-h light-dark cycle) micro-insulators. Unless otherwise specified, experiments were conducted using 3-month-old Balb/cJ male mice, purchased from Jackson Laboratories (Strain#:000651). To study the effects of glucocorticoids on bone biology, Methylprednisolone 5 mg 60-day release pellets or vehicle (cat# SG-241, Innovative Research of America, Sarasota, FL, USA) were implanted under the skin near the lumbar spine at day 0, and pellets were maintained for 56 days or removed after 28 days as indicated. Human parathyroid hormone 1-34 (hPTH 1-34) was purchased from Sigma (cat# P3796), and the treatment was administered subcutaneously at 40 µg/kg, 5x/week for four consecutive weeks. For 28-day studies with concomitant hPTH (40 µg/kg/day, subcutaneously 5x/week) or aBSG (1 mg/kg, i.v. 3x/week) treatment mice were implanted with Methylprednisolone 2.5 mg 21-day release pellets or vehicle control pellets. Aged (24-months) male and female C57BL/6 from were ordered from NIA and treated i.v. with aBSG or IgG controls (1 mg/kg, 3x/week for 4 weeks; ThermoFisher, cat#: 16-1471-82 and 16-4321-82). For renal capsule transplantation

experiments, cells from 3-month-old male GFP reporter mice (C57BL/6-Tg(CAG-EGFP)1Osb/J; JAX: 003291) were transplanted into male B6 mice (C57BL/Ka-Thy1.1-CD45.1; JAX: 000406). On the same day, mice were transplanted with GC pellets (2.5 mg 21-day release pellets or vehicle) and hPTH treatment was initiated (40 µg/kg/day, 5x/week for 3 weeks). Three-month-old immunodeficient NSG mice (NOD.Cg-Prkdcscid Il2rgtm1Wjl/SzJ; JAX: 005557) were used for human cell transplantation studies. Bsg$^{flox/flox}$ mice were obtained from the lab of Dr. Romana Nowak at the University of Illinois at Urbana-Champaign and as previously published[39]. Tamoxifen inducible Col2a1-Cre$^{ERT2}$ mice were crossed with Bsg$^{flox/flox}$ mice to generate Col2a1-Cre$^{ERT2}$ x Bsg$^{flox/+}$ (conditional heterozygous Basigin knockout, cKO) female offsprings. At 8 weeks of age, induction of the Cre-recombinase was achieved by daily oral gavage of tamoxifen (75 mg/kg) for 5 consecutive days. For the control group, transgenic mice were injected with vehicle (corn oil). One week after the first dose, mice were implanted with 21-day Methylprednisolone release pellets.

### Human primary cells and cell lines
Human skeletal stem cells (hSSCs) were obtained from fracture callus (day 3–5 after injury) tissues during open reduction and internal fixation procedures. Procurement and handling were in accordance with the guidelines set by the Stanford University Institutional Review Board (IRB-35711) and the UC Davis Institutional Review Board (IRB-1997852). Informed consent was not required as samples were considered biological waste. No restrictions were made regarding the race, gender, or age of the specimen's donor. Following excision, all specimens were placed on ice, and hSSCs were isolated as described below. Early passage (3–5) VeraVec HUVEC cells were used for endothelial assays as further explained below.

### Micro-computed tomography and dual-energy x-ray absorptiometry (DEXA) analysis
Soft tissue-free femurs were scanned within 2 h of dissection using a Bruker Skyscan 1276 (Bruker Preclinical Imaging) with a source voltage of 85 kV, a source current of 200 µA, a filter setting of Al 1 mm, and a pixel size of 17.5 µm at 2016 × 1344. Reconstructed samples were analyzed using CT Analyser (CTan) v.1.17.7.2 and CTvox v.3.3.0 software (Bruker). Anatomical landmarks as per ASBMR guidelines[40] were used to set the region of interest for analyzing trabecular (200 consecutive sections) and cortical (100 consecutive sections) bone parameters. Subsequent mechanical strength testing was conducted using 3-point bending. The maximum load (N) sustained prior to fracture was recorded[41]. Whole-body DEXA imaging was performed at baseline (day of surgery) and 14 days post-surgery to determine lumbar spine (L5) BMD. Mice were anesthetized with isoflurane and placed in a cabinet x-ray system (Mozart®, Kubtec Medical Imaging) for analysis.

### Histology
Soft-tissue-free specimens were fixed in 4% PFA at 4 °C overnight. Samples were decalcified in 400 mM EDTA (EDTA) in PBS (pH 7.2) at 4 °C for 2 weeks with a change of EDTA every other day. The specimens were then dehydrated in 30% sucrose at 4 °C overnight. Specimens were embedded in optimal cutting temperature compound (OCT) and sectioned at 5 µm. Representative sections were stained with freshly prepared hematoxylin and eosin (H&E) or Tartrate-resistant acid phosphatase (TRAP). Immunofluorescence on sections of cryopreserved long bone and ectopic bone specimens was incubated with 3% Bovine Serum Albumin in Tris Buffered Saline (TBS) for 1 h. Then, samples were probed with primary antibody (Endomucin 1:200, cat# sc-65495, Santa Cruz; CD31 1:200: cat# AF3628, ThermoFisher; Basigin 1:200, cat# 16-1471-82, ThermoFisher and cat# NB500-430, Novus Biologicals) diluted in 1% BSA/PBS and incubated in a humidified chamber at 4 °C overnight. The specimens were washed with PBS three

times. Secondary antibody (AF488 dk anti-rat 1:500: Donkey anti-Rat IgG (H + L) Highly Cross-Adsorbed Secondary Antibody, Alexa Fluor™ 488, cat# A21208; Donkey anti-Goat IgG (H + L) 1:500 Cross-Adsorbed Secondary Antibody, Alexa Fluor™ 647, cat# A21447, both Thermo-Fisher) was applied for 15 min at room temperature in the dark. Specimens were also incubated with 1 µg/mL of DAPI for 10 min and then washed twice. The specimens were then mounted with a coverslip using Fluoromount-G and imaged. ImageJ (http://wsr.imagej.net/distros/osx/ij152-osx-java8.zip; RRID:SCR_003070) was used to quantify Endomucin-positive blood vessel/sinusoid lumen area. H&E stains were analyzed using image deconvolution according to Landini et al.[42]. The bone marrow area spanning 1 mm below the growth plate between the cortical bone region was selected, and the image deconvolution plugin, Colour Deconvolution 2, was run on the selected region. The H&E 2 filter within the plug-in was applied. The color threshold on panel "Colour_2" was adjusted to represent the Eosin-stained bone regions. To measure the BV/TV, the partial (bone) area and total area were analyzed.

## Bone histomorphometry analysis

Mice were injected with 20 mg/kg of Calcein (Sigma-Aldrich, St. Louis, MO, USA) 8 and 2 days before euthanasia. Histological sections of bones were prepared as described above. A standard sampling site in the secondary spongiosa of the distal metaphysis was established. Mineral apposition rate (MAR), bone formation rate (BFR) and osteoclast number/bone surface (Oc/BS) were calculated.

## Flow cytometric isolation of skeletal progenitor cells

SSC lineage populations were isolated using cell-surface-marker profiles as previously described[10,12]. In brief, femurs were dissected, cleaned of soft tissue and crushed using a mortar and pestle. Then, the tissue was digested in M199 (cat# 11150067, ThermoFisher Scientific) with 2.2 mg/mL collagenase II buffer (cat# C6885, Sigma-Aldrich) at 37 °C for 60 min. Dissociated cells were strained through a 100-µm nylon filter, washed in staining medium (10% fetal bovine serum (FBS) in PBS) and pelleted at 200 g at 4 °C. The cell pellet was resuspended in staining medium, and red blood cells were depleted via ACK lysis for 5 min. The cells were washed again in staining medium and pelleted at 200 g at 4 °C. Then, the cells were prepared for flow cytometry with fluorochrome-conjugated antibodies. For mouse SSC lineages: CD90.1 (1:200, ThermoFisher, 47–0900), CD90.2 (1:200, ThermoFisher, 47–0902), CD105 (1:200, ThermoFisher, 13–1051), CD51 (1:200, BD Biosciences, 551187), CD45 (1:400, BioLegend, 103110), Ter119 (1:400, ThermoFisher, 15–5921), Tie2 (1:100, ThermoFisher, 14–5987), 6C3 (1:200, BioLegend, 108312), streptavidin PE-Cy7 (1:400, ThermoFisher, 25–4317), Sca-1 (1:200, ThermoFisher, 56-5981), CD45 (1:400, ThermoFisher, 11-0451), CD31 (1:200, ThermoFisher, 12-0311) and CD24 (1:200, ThermoFisher, 47–0242). For human SSC lineages: CD45 (1:200, BioLegend, 304029), CD235a (1:200, BioLegend, 306612), CD31 (1:200, ThermoFisher Scientific, 13-0319), CD202b (TIE-2) (1:100, Bio-Legend, 334204), streptavidin APC-AlexaFlour750 (1:400, Thermo-Fisher, SA1027), CD146 (1:200, BioLegend, 342010), PDPN (1:200, ThermoFisher Scientific, 17-9381), CD164 (1:200, BioLegend, 324808) and CD73 (1:200, BioLegend, 344016). Flow cytometry was conducted on a FACS Aria II Instrument (BD BioSciences) using a 70-µm nozzle in the Shared FACS Facility in the Lokey Stem Cell Institute (Stanford). The skeletal stem-cell lineage gating strategy was determined using appropriate isotype and fluorescence-minus-one controls. Propidium iodide staining was used to determine cell viability. Cells were sorted for purity. Flow cytometric analysis was conducted using FlowJo (FLOWJ LLC, v10.10). Blood panels and bone marrow analysis were conducted on a CYTEK Aurora Sorter. Blood was stained with Ter119-PE-Cy5 (1:400, 116210, BioLegend), CD45-FITC (1:400, 11-0451, Invitrogen), B220-APC-Cy7 (1:200, 103224, BioLegend), CD11b-PE-Cy7 (1:200, 101216, BioLegend), CD3-APC (1:200, 100236, BioLegend),

Gr1-BV711 (1:200, 108443, BioLegend). CD11b+ and Gr1+ cells within CD45+ were considered myeloid lineage, while CD45 + CD11b-Gr-CD3+ and B220+ were marked as lymphoid lineage. HSC lineage populations were stained with Lineage cocktail-Pacific Blue (1:50, 133310, BioLegend), CD127-BV711 (1:200, 135035, BioLegend), CD117-APC-Cy7 (1:200, 105826, BioLegend), Sca1-APC (1:200, 160904, BioLegend), CD16/32-PE (1:200, 156606, BioLegend), CD34-BV786 (1:100, 742971, BD Biosciences), CD135-PE-Cy5 (1:200, 135312, BioLegend), CD150-BV510 (1:200, 115929, BioLegend). DAPI was used as live stain.

## Mouse cell culture

Cells were cultured in α-MEM with 10% FBS and 1% penicillin–streptomycin (ThermoFisher Scientific; 15140-122). For in vitro osteogenic and chondrogenic differentiation assays, 12,000 FACS-purified mouse SSCs were cultured to sub-confluency in expansion media. Cells were washed in PBS, trypsinized and transferred to a 96-well plate with osteogenic differentiation medium containing 10% FBS, 100 µg/mL ascorbic acid and 10 mM β-glycerophosphate in α-MEM for 14 days. Alternatively, for chondrogenic differentiation micromass cultures were generated by a 5-µl droplet of cell suspension with approximately $1.5 × 10^7$ cells per mL pipetted in the center of a 24-well plate and cultured for 2 h in the incubator before adding warm chondrogenic medium consisting of α-MEM (high glucose) with 10% FBS, 100 nM dexamethasone, 1 µM L-ascorbic acid-2-phosphate and 10 ng/mL TGF-β1 (Invitrogen, cat# PHG9204). The micromass was maintained for 21 days with media changes every other day. At the end of differentiation, cells were washed in PBS, fixed in 4% PFA and stained with Alizarin Red S (cat# A5533-25G, Sigma-Aldrich) solution (osteogenesis) or Alcian Blue (cat# A3157, Sigma-Aldrich) solution (chondrogenesis). For osteogenic potential, Alizarin Red S stain was dissolved in 300 µL of 20% Methanol/10% Acetic Acid solution. After complete liberation of staining, 80 µL of each well was transferred into a new 96-well plate in duplicates and absorbance was measured at 450 nm. Chondrogenic potential was assessed by spectrophotometrically measuring absorption at 595 nm. For osteoclastogenesis assays flushed bone marrow cells from long bones were plated in 24-well plates at a density of 200,000 cells per well with α-MEM without phenol red, 1% GlutaMAX supplement (cat# 35050061, Gibco), 10% FBS, 1% Penicillin-Streptomycin 10,000 U/mL, 1 µM prostaglandin E2 (cat# P0409, Sigma), and 10 ng/mL Csf1 recombinant murine protein (cat# 315-02, Peprotech) for 3 days. Starting on day three, the media was changed daily to also include 10 ng/mL recombinant mouse RANKL (cat# 315-11, Peprotech). Osteoclast culture continued for 10 days until large, multinucleated osteoclasts appeared. Plates were stained for osteoclasts using the TRAP kit (cat# 387A, Sigma-Aldrich).

## Human SSC (hSSC) culture

Freshly sorted, primary hSSCs were cultured in α-MEM with 10% human platelet-derived lysate (HPL; cat# 06962, STEMCELL technologies), 1% Penicillin-Streptomycin, 0.01% heparin and maintained at 37 °C with 5% $CO_2$. For in vitro differentiation assays, hSSCs were treated and supplemented with the same osteogenic and chondrogenic differentiation protocols as described for mouse cells above. For viral overexpression experiments, the coding region of Basigin (269aa) was cloned into a pCCLc backbone for manufacture of second-generation lentivirus using Lenti-X 293T as packaging cells, as previously described[43]. The viral titer was determined functionally, based on the amount of virus necessary to reach 90–95% eGFP expression. Cells were transduced using Lipofectamin3000 (cat# L3000001, ThermoFisher). For rescue experiments media was either supplemented with hPTH 1-34 by adding a final concentration of 2.5 nM 6 h before each media change (to resemble intermittent exposure) or with 1 µg/mL of monoclonal Basigin antibody (cat# NB55-430, Novus Biologicals) during media change. Reactive oxygen species measurement

was performed according to the manufacturer's instructions (cat# ab113851, Abcam).

## Cell line experiments

Low passage VeraVec endothelial cells were thawed in a 10 cm dish with 10 mL of EGM-2 medium (cat# CC-3156, Lonza Bioscience). VeraVecs were grown at 37 °C with 5% $CO_2$ until confluent, with media changes every 3 days. Before the assays, VeraVec culture was starved with low serum medium. For migration (wound scratch assay)[44], VeraVecs were lifted with 0.05% trypsin, spun down and resuspended in supernatant from 48 h cultured SSCs. They were then plated at $1.5 \times 10^5$ cells/well in a 24-well plate. To induce a gap for the migration, a scratch assay was generated using a 1000 μL micropipette tip, scratching vertically from one side of the well to the other. The gap was imaged, and area measured at times 0 h and 12 h. The area healed was calculated by (area at 0 h − area at 12 h)/(area at 0 h) × 100%. For tube formation, a matrix (cat# A1413201, Geltrex, ThermoFisher) was deposited in wells of a 48-well plate at an amount of 50 μL/cm², for a total of 50 μL per well. As for migration, VeraVecs were lifted and resuspended in supernatant from SSCs. VeraVecs were plated at $1.5 \times 10^4$ cells/well. Cells were imaged at 12 h.

## Subcutaneous xenografts

Primary hSSCs were transplanted into the dorsum of 3-month-old immunodeficient NSG mice (NOD.Cg-Prkdcscid Il2rgtm1Wjl/SzJ; JAX: 005557) as described previously[45]. Briefly, freshly sorted patient-derived hSSCs were sorted, expanded to confluency and virally transduced to either overexpress Basigin or control. $1 \times 10^6$ cells were mixed with 5 μL Matrigel and seeded on macroporous composite scaffolds formed of hydroxyapatite (HA) and poly(lactide-co-glycolide) (PLG) (HA-PLG) on ice. Scaffolds were fabricated using a gas foaming/particulate leaching method as previously described[46]. Microspheres composed of PLG (85:15; DLG 7E; Lakeshore Biomaterials, Birmingham, AL) were prepared using a double-emulsion process and lyophilized. Lyophilized microspheres (7.1 mg) were combined with 17.8 mg of synthetic HA (particle size, <200 nm; Aldrich Chemistry, St. Louis, MO) and 134.9 mg of NaCl (300 to 500 μm in diameter) to yield a 2.5:1:19 mass ratio of ceramic:polymer:salt. The powdered mixture was compressed under 2 metric tons for 1 min to form solid disks (final dimensions, 8 mm in diameter and 1.5 mm in height) using a Carver Press (Carver Inc., Wabash, IN). Compressed disks were exposed to high-pressure $CO_2$ gas (5.5 MPa) for at least 24 h, followed by rapid pressure release to prompt polymer fusion. Salt particles were leached from scaffolds in distilled $H_2O$ for 24 h to generate HA/PLG composite scaffolds. HA/PLG composite scaffolds were cut with a biopsy punch to produce scaffolds with final dimensions of 4 mm in diameter and 1.5 mm in height. Scaffolds were sterilized in 70% ethanol in 24-well plates for 20 min, followed by two rinses in sterile PBS. Sterile scaffolds were dried and kept until use. To promote cell adhesion, scaffolds were incubated in culture media at 37 °C with 5% $CO_2$ for 30 min directly before cell seeding. A small skin incision was made in the dorsum of anesthetized NSG mice, and the cell containing scaffold was slid under the skin. Interrupted sutures were applied to close the incision, and transplants were allowed to engraft and grow for 6 weeks.

## Renal capsule transplantation

Renal capsule transplantations were conducted as previously described[10,12]. Briefly, in the anaesthetized mouse, a 5-mm dorsal incision was made, and the kidney was exposed manually. Then, a 2-mm incision was created in the renal capsule using a needle bevel, and 5000 FACS-purified mouse SSCs resuspended in 2 μl of Matrigel were transplanted beneath the capsule. The kidney was re-approximated manually, and incisions were closed using sutures and staples. At the time of surgery, mice were randomly divided into four groups. Mice either received subcutaneous placebo, hPTH or Methylprednisolone pellets as described above. One additional Methylprednisolone pellet group was treated with hPTH (see above). Grafts were collected after 21 days.

## Single-cell RNA-sequencing

Femurs were collected from all the treatment groups, and each was processed, digested and prepared for FACS as described above. Single cell solutions of each treatment group were then pooled ($n = 5$ per group), and $1 \times 10^6$ PI-Ter119-CD45+ and $1 \times 10^6$ PI-Ter119-CD45- cells were sorted into one collection tube containing FACS buffer for each experimental group. Cells were then processed with 10X Chromium Next GEM Single Cell 3′ GEM kit (10X Genomics, v.3.1) according to the manufacturer's instructions to target 5000 cells per group, sequenced on a partial lane Illumina NovaSeq platform. Barcoded samples were demultiplexed, aligned to the mouse genome (GRCm39.104), and UMI-collapsed with the Cellranger toolkit with standard settings (v.7.1.0, 10X Genomics). We used the Scanpy package (v.1.9.1.) to explore the data. First, quality filtering to exclude multiplets and cells of poor quality was conducted by only keeping cells with a gene count of more than 250 and fewer than 3000, with less than 15% mitochondrial and 15% ribosomal gene content, leaving 5735 cells. Genes expressed in fewer than three cells across all cells were also removed from downstream analysis. Data were log-normalized, cell cycle-regressed and scaled for analysis. Dimensionality reduction and Leiden clustering, as well as subclustering, were conducted, choosing parameters based on PCA elbow plots. For single-cell RNA-sequencing of SSC-derived grafts, the tissue was dissected out and processed as described for femur bone processing to isolate SSCs above. Single cell solution was then stained with PI and Ter119, and living (PI-negative), non-red blood cells (Ter119-negative) were sorted into FACS buffer for each group. Based on yield, cells were then processed with 10X Chromium Next GEM Single Cell 3′ GEM kit (10X Genomics, v.3.1) according to the manufacturer's instructions to target 1000 cells for Placebo, hPTH and GC + hPTH groups, as well as 500 cells for GC group. All other steps were conducted as described above. A total of 852 graft-derived cells across all groups passed stringent quality filtering. Data are available under GEO accession number GSE253044.

## Statistical analysis

Statistical significance between placebo and treatment groups was determined using two-tailed, unpaired Student's t-test or One-way ANOVA for multiple groups comparison with LSD Fisher test unless stated otherwise in the figure legend (GraphPad Prism; version 10). Statistical significance was defined as $p < 0.05$. All data points refer to biological replicates and are presented as mean ± standard error of the mean (SEM) unless otherwise stated in the figure legend.

## Reporting summary

Further information on research design is available in the Nature Portfolio Reporting Summary linked to this article.

# Data availability

Single cell RNA-sequencing data was deposited in the GEO database with access number GSE253044. Source data are provided with this paper.

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

## Acknowledgements

We thank Stanford University Institute for Stem Cell Biology and Regenerative Medicine FACS core (NIH S10 RR02933801) for experimental support, and NIH S10 1S10OD028493-01A1 (Principal Investigator: Charles K. F. Chan). We are grateful to Dr. Romana Nowak at the University of Illinois at Urbana-Champaign for generously providing Basigin^flox/flox mice for this study. MicroCT work on the Bruker Skyscan 1276 was supported by NIH S10 Shared Instrumentation Grant (1S10OD02349701, PI Timothy C. Doyle). Further funding for this work was provided by NIH–NIA K99/R00AG066963 to T.A., NIH–NIA K99/R00AG049958, Stanford Wu Tsai Human Performance Alliance Funds, Siebel Foundation, the Heritage Medical Foundation, Prostate Cancer Foundation, the American Federation for Aging Research (AFAR)–Arthritis National Research Foundation (ANRF), a seed grant from the WHSDM Stanford Women's Health and Sex Differences in Medicine Center, and an endowment from the DiGenova Family to C.C. D.M. was supported by a National Institute of Arthritis and Musculoskeletal and Skin Diseases funded T32 training program in Musculoskeletal Health Research (T32 AR079099). E.W. and K.W. were supported by the California Institute of Regenerative Medicine (CIRM) EDUC4-12792 Research Training Program. A.M. and F.C. were supported by CIRM grant EDUC2-12691. Funding to N.L. was provided by NIH-NIA R01AG070647 and the Research and Education Fund of UC Davis Health.

## Author contributions

N.L. conceived the work, supervised the study and wrote the manuscript. T.A. co-conceived the study, designed experiments, performed experiments, co-supervised the study and wrote the manuscript. C.C. co-supervised the study and helped design experiments. D.M. and K.C. performed the majority of the experiments and analyses. E.H., K.W., A.M., F.C., Y.W., L.Z., L.W., M.M., E.W., A.C., A.S., K.L., and F.F. contributed expertise, performed experiments and/or data analysis.

## Competing interests

The authors declare no competing interests.
