## [Transparent Peer Review file · Nature Communications]

Basigin Links Altered Skeletal Stem Cell Lineage Dynamics with Glucocorticoid-induced Bone Loss and Impaired Angiogenesis

Corresponding Author: Dr Thomas Ambrosi

Version 0:

Reviewer comments:

Reviewer #1

(Remarks to the Author)

The manuscript by Ambrosi et al. reported an elevated Basigin expression from SSCs upon GC treatments and established a connection between this phenomenon and GC-induced bone loss. Also, they found that the anti-Basigin treatments reversed the detrimental effects of GC in bone and thus may be a promising new therapy for GC or age-induced bone loss. The experimental design is overall logical, and the data are presented clearly. But several points need to be addressed:

1. The lack of GCrem+PTH group is a miss of opportunity.
2. Fig 1h, the smooth muscle cells are completely gone in the GC rem group but very high in the GC group. Please explain more on this.
3. The function of PTH in angiogenesis was reported in several publications and is not in line with the data reported in Fig F. Discussion is needed.
4. Fig 3. g and h showed that PTH treatment has more senescence markers than placebo. Discussion is needed.
5. Fig 5C, the PTH treatment rescued Bsg-OE, suggesting they may regulate the same downstream effector(s). It would be worthwhile to explore this further or have a sufficient discussion.
6. Fig.7a. Using histomorphometry to show the in vivo effect of aBSG treatment on osteoblast and osteoclast parameters in aging mice would provide more insights into how such treatment improves aging bone. In particular, the ex vivo pro-osteoclastogenic effect of aBSGs is inconsistent with the observation in Fig. 6d and the final schematics for the mechanism.
7. It is unclear how Basigin influences SSC function and impairs osteogenesis and chondrogenesis as shown in Figures 6 and 7. Some sequencing/bioinformatics experiments, similar to those in Fig 1 and 3, could provide understanding of this important aspect of Basigin function and support the aBSG therapy for aging- or GC-induced bone loss.

Reviewer #2

(Remarks to the Author)

In this manuscript, the authors proposed SSC-derived Basigin may contribute to glucocorticoid-induced bone loss. Since the role of Basigin in bone pathophysiology has never been reported, the authors' findings on Basigin are potentially interesting.

However, this manuscript has several critical concerns. Importantly, the authors failed to provide genetic evidence for the pathological significance of Basigin in glucocorticoid-induced bone loss. The cellular target or mode of action of Basigin remain also unclear.

Taken together, this manuscript is not suitable for publication in Nature Communications in its current form.

1) The importance of Basigin and its receptor, and the cellular sources and targets of Basigin should be confirmed by conditional KO experiments.

2) The precise molecular mechanisms underlying the GC-induced Basigin expression in SSCs and Basigin downstream signaling in target cells should be clarified.

Reviewer #3

(Remarks to the Author)

In this study, Ambrosi et al. proposed that Basigin plays a fundamental role in GC-induced bone loss and impaired angiogenesis. Specifically, SSC-derived Basigin inhibits endothelial cell migration and network formation, which could contribute to osteonecrosis. Importantly, antibody-mediated depletion of Basigin alleviates GC's detrimental effects on osteogenesis and angiogenesis, which could be used to treat osteoporosis in aged mice. Overall, this is an interesting report on a potential new target to treat GC-induced osteoporosis and osteonecrosis, which uncovered a novel molecular link between SSC-EC crosstalk. However, several concerns need to be addressed in a potential revision.

1. The molecular mechanisms by which Basigin inhibits endothelial cell function are unclear. Whether Basigin functions as coreceptor for Kdr/Vegfr2 in bone marrow endothelial cells, and what's the downstream signaling events that leads to impair angiogenesis should be elucidated.
2. Mouse SSCs have been previously defined as CD51+CD200+CD105- on top of other negative lineage markers by Chan et al. (Cell, 2015). In this study, the authors did not include the critical positive marker CD200 (lines 148-149), rendering loose definition of SSCs. Since LepR+ bone marrow stromal cells (BMSCs) are also CD51+, the SSC population analyzed in this study could largely overlap with LepR+ BMSCs, instead of mouse SSCs identified by Chan et al. Similarly, the phenotypic markers that define SSPCs (line 192) were not specified. Therefore, the authors should provide more detailed description and explanation of the populations they analyzed to avoid confusion.
3. For the scRNA-seq experiments, they did not perform any enrichment of the non-hematopoietic populations after digesting the bone marrow, which could cause overrepresentation of blood cells (>95%) as compared to BMSCs (0.3%) and endothelial cells (0.1%). However, this is not the case as shown in Figure 1e. How do they explain this apparent discrepancy?
4. Genetic deletion data for Basigin were lacking. Basigin could also be synthesized by immune cells to promote systemic or local inflammation. To prove that SSC-derived Basigin caused bone marrow angiogenic defects, one should perform conditional deletion of Basigin in SSCs together with GC administration.
5. Since GC has been widely used as an anti-inflammatory drug, why does it promote inflammation, as well as osteoclastogenesis and bone resorption, in the bone marrow by up-regulating Basigin? This seems paradoxical. More legitimate discussions should be provided on this point.
6. Figure 4d should be replaced by images with clearer cell morphology.

Version 1:

Reviewer comments:

Reviewer #1

(Remarks to the Author)

The questions are answered appropriately and sufficiently. No more questions, and I recommend an acceptance.

Reviewer #2

(Remarks to the Author)

Thank you for addressing all of my comments. The addition of the cKO data has significantly improved the manuscript, and I have no further comments.

Reviewer #3

(Remarks to the Author)

The authors have addressed most of my concerns. However, for point #2, what does it mean that CD200 is "redundant for this study"? They have defined SSCs using one single positive marker CD51 (CD45-Ter119-Tie2-CD90-6c3-CD105-CD51+, line 149-150), which is inconsistent with the mSSC definition in the Chan study (Cell, 2015, CD45-Ter119-Tie2-CD90-6c3-CD105-CD51+, reference #10). CD51 is not only expressed in mSSCs and LepR+ BMSCs, but also highly expressed in osteoprogenitors, pre-osteoblasts and mature osteoblasts. Therefore, without adding CD200 as an additional positive marker, any conclusion about SSCs in this study becomes shaky. Please avoid overclaim on SSCs throughout the manuscript if they actually analyzed a rather heterogeneous population.

Version 2:

Reviewer comments:

Reviewer #3

(Remarks to the Author)

No further comments.

Basigin Links Altered Skeletal Stem Cell Lineage Dynamics with Glucocorticoid-induced Bone Loss and Impaired Angiogenesis (Nature Communications manuscript NCOMMS-24-47461)

Responses to	Comments	Page
Reviewer 1	1-7	2-6
Reviewer 2	1-2	7-8
Reviewer 3	1-6	9-11

Changes to the text were highlighted in yellow. New and substantially updated figure panels were similarly highlighted that way.

Introduction to revision:

Dear Reviewers,

We would like to thank you for your helpful and constructive feedback. We were happy to see that all reviewers generally shared our enthusiasm about the novel findings of this study. In our point-by-point response below you will see that we have addressed all reviewer comments by adding clarifications, a more detailed discussion and conducting new experiments bolstering our claims. The substantial new data supports our main conclusions, thereby further strengthening our initial manuscript. In particular:

1. We have generated and analyzed skeletal stem and progenitor cell-specific Basigin knockout mice showing that heterozygous loss of Basigin in the mesenchymal lineage is sufficient to prevent GC-induced bone loss.
2. We have added female mice to our aging experiments showing that aged female mice treated with monoclonal antibody against Basigin for four weeks improve bone mass as previously demonstrated in male mice.
3. While the molecular mechanisms facilitated by Basigin are pleiotropic, we provide a more detailed discussion on its role in the newly discovered SSC-endothelial signaling axis based on existing literature, functional findings and transcriptional readouts. In any case, the identification of Basigin as a prominent factor in GIOP as well as the elucidation of bone-forming skeletal stem cells as the source of Basigin driving bone loss in conjunction with vascular interactions and the therapeutic implications of targeting it, should be considered novel and highly valuable for the scientific community. Due to the scope of the present study, timeline and current funding landscape we were not able to conduct extensive experiments needed to do a thorough investigation into the mechanistic underpinnings of Basigin's molecular actions in SSCs and endothelial cells. Even though it is of great interest, this will be the focus of follow-up work.

Please find our point-by-point responses below. We would like to thank all reviewers again and hope that they consider the updated manuscript acceptable.

Sincerely,

Drs. Ambrosi and Lane for the authors.

Point-by-Point Response

REVIEWER COMMENTS

Reviewer #1 (Remarks to the Author):

The manuscript by Ambrosi et al. reported an elevated Basigin expression from SSCs upon GC treatments and established a connection between this phenomenon and GC-induced bone loss. Also, they found that the anti-Basigin treatments reversed the detrimental effects of GC in bone and thus may be a promising new therapy for GC or age-induced bone loss.

The experimental design is overall logical, and the data are presented clearly. But several points need to be addressed:

We thank the reviewer for the positive and thorough feedback. Please find our response to all points raised below.

1. The lack of GCrem+PTH group is a miss of opportunity.

Response: We appreciate the reviewer's observation and agree that a GCrem+PTH group would have been a logical addition. For this study we built on previous work, where we demonstrated a rescue of skeletal changes when animals exposed to glucocorticoids were concurrently treated with hPTH (PMID: 18975341). However, we indeed conducted an initial experiment with the additional group that received hPTH after glucocorticoid removal. As you can see below, in contrast to continuous hPTH treatment during glucocorticoid exposure, treating mice with hPTH right after discontinuation of glucocorticoids did not rescue any of the skeletal phenotypes induced by glucocorticoids. Due to these unexpected results, we decided not to perform additional experiments (e.g., single cell RNA sequencing) for this group and to build on a treatment group that was able to prevent glucocorticoid induced bone loss (and showed corresponding functional readouts – **Fig.2**). We believe that including the additional hPTH group in the manuscript would have added too much confusion, required substantial additional discussions as well as experiments and distracted from the main focus of this study.

Inbred mice can present with significant variability in bone mass between individual age-matched animals (PMID: 8739896). In previous studies we determined that bone mass changes can be difficult to determine between treatment groups when a baseline measurement was not obtained. By scanning the experimental animals at baseline and then at the end point of the study we can find significant differences of skeletal parameters in studies with mice exposed to glucocorticoids and different treatment groups (PMID: 18975341).

Unfortunately, we were not able to do the baseline scan for each individual mouse in this pilot study which might have contributed to the unexpected results shown below. Instead, we conducted baseline measurements of age-matched control mice. The reason behind that approach was that we were specifically interested in the impact of skeletal stem cell-based processes in response to glucocorticoid treatment. We previously showed that even short radiation exposure can significantly impair skeletal stem cell function in vitro and in vivo (PMID: 25594184, PMID: 34280086). A separate study will have to explore the unexpected observations by performing microCT scans of individual animals of all experimental groups at baseline and at the end of the study to determine the responses of single mice more closely. This was beyond the scope of the current manuscript that sought to investigate a new strategy that could prevent GIOP.

Point-by-Point Response

2. Fig 1h, the smooth muscle cells are completely gone in the GC rem group but very high in the GC group. Please explain more on this.

Response: Thank you for this comment. Basigin, which is upregulated with GC is a known activator of smooth muscle cell proliferation (PMID: 25149188). GC removal and hPTH treatment reduce Basigin levels which might diminish this cell population as observed. Even though fitting with the overall results of our study but given the low abundance of Smooth muscle cells in the scRNAseq dataset any conclusions should be taken with a grain of salt. Consequently, we decided to remove any comments on smooth muscle cells in scRNAseq data analysis from the updated manuscript.

3. The function of PTH in angiogenesis was reported in several publications and is not in line with the data reported in Fig F. Discussion is needed.

Response: We would like to emphasize that blood vessel phenotypes are the response to GC exposure or GC with hPTH, not hPTH alone. Given our observations indicate that vascular alterations are secondary to changes of the bone-forming cells which benefit from hPTH directly, this might explain discrepancies with previous studies focusing on hPTH specifically. In any case, hPTH's role in angiogenesis is complex and context dependent. Direct beneficial actions on endothelial cells have been reported in fracture regeneration settings (PMID: 35269519, PMID: 31821371) which might also benefit blood vessel architecture in GC exposed mice. Therefore, our findings are not at odds with previous publications. This was also addressed in the discussion (II.383-391).

4. Fig 3. g and h showed that PTH treatment has more senescence markers than placebo. Discussion is needed.

Response: Thank you for this comment. Senescence markers commonly overlap with markers of cellular stress/pro-inflammatory states. There is ample evidence that higher circulating hPTH levels can drive a pro-inflammatory state and oxidative stress (PMCID: PMC3980926; PMID: 30662585). On the other hand, hPTH can be protective of stress and senescence (PMCID: PMC8089505. PMID: 29956812). Slight increases in those

Point-by-Point Response

markers might just be the reflection of a more activated state of hPTH exposed cells compared to placebo group. In the context of GC exposure, which promotes bone marrow inflammatory signaling in part through higher osteoclastogenesis, hPTH seems to exert the described protective role. The text has been amended to address this observation (II.209-212).

5. Fig 5C, the PTH treatment rescued Bsg-OE, suggesting they may regulate the same downstream effector(s). It would be worthwhile to explore this further or have a sufficient discussion.

Response: We show that Basigin drives accumulation of SSCs and increases colony forming ability at the expense of differentiation in vitro when overexpressed in human SSCs. Blockade of Basigin seems to promote SSC differentiation rather than proliferation, similar to what is achieved with hPTH treatment. We agree that this might than allow the same proper cellular interactions to play out, but in the absence of detailed mechanistic studies, which are beyond the scope of this manuscript, we cannot rule out distinct downstream effectors. We have added this to the discussion (II.402-405).

6. Fig.7a. Using histomorphometry to show the in vivo effect of aBSG treatment on osteoblast and osteoclast parameters in aging mice would provide more insights into how such treatment improves aging bone. In particular, the ex vivo pro-osteoclastogenic effect of aBSGs is inconsistent with the observation in Fig. 6d and the final schematics for the mechanism.

Response: Thank you for this comment. For the aging studies we were not able to include dynamic histomorphometry since the labeling would have interfered with flow cytometric analyses. However, in the updated manuscript we show for both aged female (expressing high levels of Basigin in the bone marrow space) and male mice, bone parameters are improved with aBSG treatment despite higher TRAP+ osteoclasts numbers as determined by in situ staining (**Fig.7a-f**). This might be a reflection of increased bone remodeling activity initiated by the monoclonal antibody treatment leading to higher stem cell-based bone formation (see in vitro assay, **Fig.7e**) initiating osteoclast activity through expression of factors like RANKL in conjunction with an increased pool of osteoclast progenitors in the aged bone marrow (PMID: 34381212) which is not present in young GC treated mice. As shown by our analyses, bone parameters improve with aBSG despite increased osteoclast activity independent of sex. We have updated the schematic accordingly by focusing on the GIOP mechanism only.

Point-by-Point Response

7. It is unclear how Basigin influences SSC function and impairs osteogenesis and chondrogenesis as shown in Figures 6 and 7. Some sequencing/bioinformatics experiments, similar to those in Fig 1 and 3, could provide understanding of this important aspect of Basigin function and support the aBSG therapy for aging- or GC-induced bone loss.

Response: Thank you for this comment. This study identifies and establishes Basigin as an important and targetable factor in GC-induced bone loss through actions on an SSC-endothelial signaling axis which entails the large body of work presented here. In new data, we show that genetic ablation of Basigin in the skeletal lineage is sufficient to prevent GC-induced bone loss (**Fig.6j-m**). This is in support of our data showing that Basigin directly impairs SSC function. While the exact mechanism of action of Basigin is of great interest to us, we believe its elucidation is beyond the scope of the current manuscript, especially due to the pleiotropic actions that have been reported for Basigin. We have added an extensive discussion of potential mechanisms supported by gene expression data (**Fig.3 & Extended Data Fig.3e,f**) to the updated manuscript (ll.325-357). Briefly, in SSCs Basigin might facilitate increased lactate influx through monocarboxylate transporters (MCTs), driving

Point-by-Point Response

increased proliferation and preventing differentiation through an altered metabolic state (PMID: 33770122, PMID: 36921622, PMID: 39934144). Additionally, modulation of the ECM by Basigin likely also affects SSC function, for example through altered mechano-sensing (PMID: 15928045). Paracrine signaling in endothelial cells allows Basigin to act as a co-receptor for Vegfr, overactivating angiogenesis leading to enhanced cellular stress and vascular dysfunction (PMID: 25825981, PMID: 24300855, PMID: 35419169). Follow-up studies will have to functionally test these potential mechanisms that are supported by transcriptomic findings and functional readouts of the present study.

Point-by-Point Response

Reviewer #2 (Remarks to the Author):

In this manuscript, the authors proposed SSC-derived Basigin may contribute to glucocorticoid-induced bone loss. Since the role of Basigin in bone pathophysiology has never been reported, the authors' findings on Basigin are potentially interesting.

However, this manuscript has several critical concerns. Importantly, the authors failed to provide genetic evidence for the pathological significance of Basigin in glucocorticoid-induced bone loss. The cellular target or mode of action of Basigin remain also unclear.

Taken together, this manuscript is not suitable for publication in Nature Communications in its current form.

We would like to thank the reviewer for the time and comments. We now provide genetic evidence for the pathological significance of skeletal stem and progenitor derived Basigin in GIOP as suggested by the reviewer. We also include a detailed discussion of potential mechanistic processes that were implied by our results but beyond the scope of the current manuscript and that we therefore were not able to address experimentally. Please consider our responses below.

1) The importance of Basigin and its receptor, and the cellular sources and targets of Basigin should be confirmed by conditional KO experiments.

Response: In response to the reviewer's concern and relevant to our findings, we have generated mice with inducible conditional knockout of Basigin in the skeletal lineage. Since our data indicated Basigin is upregulated at the stem and progenitor level upon GC exposure we employed the Col2a1-Cre^{ERT2} mouse model. As you can see in **Fig.3d**, Col2a1 was highly specific to the SSPC cluster. Furthermore, our previously published dataset of skeletal lineage cell type specific gene expression (from PMID: 34381212) shows specific expression of Col2a1 at the multipotent state but not more committed progenitor level (see figure on right). While we acknowledge that Col2a1-Cre lines have also been used to target chondrocytes, we did not observe Basigin expression in chondrogenic tissues during any of our studies and critically in response to GC exposure. More generally, there are no specific Cre-lines that exclusively target skeletal stem and progenitor cells. In order to do these studies, we were kindly provided with

Basigin floxed mice (Bsg^{fl/fl}) by the lab of Dr. Romana Nowak (Department of Animal Sciences, University of Illinois at Urbana-Champaign, Urbana, IL, USA) who generated these mice and published on them (PMID: 34106247). After crossbreeding with Col2a1-Cre we were able to generate heterozygous knockout mice. One week after tamoxifen induction, conditional knockout mice were exposed to GCs for four weeks. In support of our previous findings, reduction of Basigin levels from the SSPC compartment through genetic ablation was sufficient to prevent GC-induced bone loss in these mice (**Fig.6j-m** & II.275-287). These findings confirm our initial findings using a genetic model and provide the basis for future investigations into the receptor and potential role of other cell sources for Basigin. Studying homozygous Basigin knockout mice will also be of interest to interrogate if complete loss of SSPC-derived Basigin might have an even more pronounced beneficial effect on GIOP protection.

Point-by-Point Response

2) The precise molecular mechanisms underlying the GC-induced Basigin expression in SSCs and Basigin downstream signaling in target cells should be clarified.

Response: As responded to comment 7 by Reviewer 1: This study identifies and establishes Basigin as an important and targetable factor in GC-induced bone loss through actions on an SSC-endothelial signaling axis which entails the large body of work presented here. In new data, we show that genetic ablation of Basigin in the skeletal lineage is sufficient to prevent GC-induced bone loss (**Fig.6j-m**). This is in support of our data showing that Basigin directly impairs SSC function. While the exact mechanism of action of Basigin is of great interest to us, we believe its elucidation is beyond the scope of the current manuscript, especially due to the pleiotropic actions that have been reported for Basigin. We have added an extensive discussion of potential mechanisms supported by gene expression data (**Fig.3 & Extended Data Fig.3e,f**) to the updated manuscript (ll.325-357). Briefly, in SSCs Basigin might facilitate increased lactate influx through monocarboxylate transporters (MCTs), driving increased proliferation and preventing differentiation through an altered metabolic state (PMID: 33770122, PMID: 36921622, PMID: 39934144). Additionally, modulation of the ECM by Basigin likely also affects SSC function, for example through altered mechano-sensing (PMID: 15928045). Paracrine signaling in endothelial cells allows Basigin to act as a co-receptor for Vegfr, overactivating angiogenesis leading to enhanced cellular stress and vascular dysfunction (PMID: 25825981, PMID: 24300855, PMID: 35419169). Follow-up studies will have to functionally test these potential mechanisms that are supported by transcriptomic findings and functional readouts of the present study.

Point-by-Point Response

Reviewer #3 (Remarks to the Author):

In this study, Ambrosi et al. proposed that Basigin plays a fundamental role in GC-induced bone loss and impaired angiogenesis. Specifically, SSC-derived Basigin inhibits endothelial cell migration and network formation, which could contribute to osteonecrosis. Importantly, antibody-mediated depletion of Basigin alleviates GC's detrimental effects on osteogenesis and angiogenesis, which could be used to treat osteoporosis in aged mice. Overall, this is an interesting report on a potential new target to treat GC-induced osteoporosis and osteonecrosis, which uncovered a novel molecular link between SSC-EC crosstalk. However, several concerns need to be addressed in a potential revision.

We appreciate the reviewer's positive feedback and suggestions which have helped to further improve our manuscript.

1. The molecular mechanisms by which Basigin inhibits endothelial cell function are unclear. Whether Basigin functions as coreceptor for Kdr/Vegfr2 in bone marrow endothelial cells, and what's the downstream signaling events that leads to impair angiogenesis should be elucidated.

Response: This study identifies and establishes Basigin as an important and targetable factor in GC-induced bone loss through actions on an SSC-endothelial signaling axis which entails the large body of work presented here. In new data, we show that genetic ablation of Basigin in the skeletal lineage is sufficient to prevent GC-induced bone loss (**Fig.6j-m**). This is in support of our data showing that Basigin directly impairs SSC function. While the exact mechanism of action of Basigin is of great interest to us, we believe its elucidation is beyond the scope of the current manuscript, especially due to the pleiotropic actions that have been reported for Basigin. We have added an extensive discussion of potential mechanisms supported by gene expression data (**Fig.3 & Extended Data Fig.3e,f**) to the updated manuscript (ll.325-357). Specifically, regarding endothelial function we conclude, ll336-347: "At the endothelial cell level, we found that GC induced vascular impairments are closely tied to SSC-derived Basigin. Supporting our findings connecting Basigin to blood vessel formation is its known role as a coreceptor for vascular endothelial growth factor receptor 2 (Kdr/Vegfr2) in endothelial cells enhancing their Vegfr mediated activation and downstream signaling. Continuous GC treatment in mice increased endothelial cell number and blood vessel area. However, Basigin exposure to endothelial cells resulted in irregularities in the connectivity and thickness of the nascent blood vessels. Flow cytometric and histological readouts, single cell gene expression data and functional experiments connect these vascular changes to enhanced Vegfr signaling with a pathological blood vessel phenotype and increased oxidative stress. This is corroborated by studies that have shown that overactivation of Vegfr signaling is a driver of cellular stress in part through the stimulation of cell damaging and inflammatory reactive oxygen species that negatively affect formation, migration and permeability of blood vessels." Follow-up studies will have to functionally test these potential mechanisms that are supported by transcriptomic findings and functional readouts of the present study.

2. Mouse SSCs have been previously defined as CD51+CD200+CD105- on top of other negative lineage markers by Chan et al. (Cell, 2015). In this study, the authors did not include the critical positive marker CD200 (lines 148-149), rendering loose definition of SSCs. Since LepR+ bone marrow stromal cells (BMSCs) are also CD51+, the SSC population analyzed in this study could largely overlap with LepR+ BMSCs, instead of mouse SSCs identified by Chan et al. Similarly, the phenotypic markers that define SSPCs (line 192) were not specified. Therefore, the authors should provide more detailed description and explanation of the populations they analyzed to avoid confusion.

Response: Thank you for the comment. We have previously shown that all SSCs in adult mice are CD200 positive (PMID: 34381212) and therefore this marker is redundant for this study. The 2015 Chan et al. study employed cells from pre-natal and early post-natal stages for functional assays. We have also demonstrated that LepR is an unspecific marker capturing both SSC and a wide spectrum of non-SSC populations (PMID:

Point-by-Point Response

34280086) making the marker combination used here a superior enrichment method for SSCs. Given discrepancies in protein and gene expression profiles as well as limitations in transcriptomic coverage of scRNAseq approaches, it is difficult if not impossible to identify highly enriched SSC clusters/population in 10X Genomics data. Because of that, we more broadly (carefully) assigned clusters as stem and progenitor cell enriched (SSPCs) rather than calling them SSCs. We have added detail in the updated manuscript as requested (ll.191-199).

3. For the scRNA-seq experiments, they did not perform any enrichment of the non-hematopoietic populations after digesting the bone marrow, which could cause overrepresentation of blood cells (>95%) as compared to BMSCs (0.3%) and endothelial cells (0.1%). However, this is not the case as shown in Figure 1e. How do they explain this apparent discrepancy?

Response: Thank you for this observation. As correctly pointed out non-hematopoietic cells are rare compared to the bone marrow resident cell types of the blood and immune system. To enrich for non-hematopoietic cells for scRNAseq analyses we sorted equal amounts of CD45+ and CD45- cells into one collection tube for each experimental group. Sorted cells were then processed for 10X Genomics scRNAseq. We apologize for this omission. We have corrected and updated the text and methods accordingly (ll.123-125 & 629-631).

4. Genetic deletion data for Basigin were lacking. Basigin could also be synthesized by immune cells to promote systemic or local inflammation. To prove that SSC-derived Basigin caused bone marrow angiogenic defects, one should perform conditional deletion of Basigin in SSCs together with GC administration.

Response: Thank you for this suggestion. Please refer to our response to comment 1 of Reviewer 2 who raised a similar concern. We have conducted this experiment using Col2a1-Cre^{ERT2} x Bsg^{fl/fl} mice. Indeed, conditional heterozygous deletion of Basigin from SSC-enriched, non-immune/non-endothelial cells was sufficient to prevent GC-induced bone loss (**Fig.6j-m**). This supports our initial conclusion that SSPC derived Basigin is the main driver of GIOP.

5. Since GC has been widely used as an anti-inflammatory drug, why does it promote inflammation, as well as osteoclastogenesis and bone resorption, in the bone marrow by up-regulating Basigin? This seems paradoxical. More legitimate discussions should be provided on this point.

Response: We thank the reviewer for this comment. We have revisited our results to more carefully assess potential “inflammation”. Pro-inflammatory markers broadly overlap with markers of cellular stress and SASP.

Point-by-Point Response

Any pro-inflammatory signaling is likely the result of indirect mechanisms related to changes in bone formation dynamics that favor increased osteoclastogenesis and pathological remodeling of blood vessel architecture that drive disruption in oxygen supply favoring pro-necrotic states at least in part through Basigin mediated mechanisms. Moreover, osteoclasts, showing increased activity with GC treatment, are myeloid-derived cells which are known inflammatory drivers (GCs also decrease Lymphoid-to-Myeloid ratio in blood – **Fig.6h**). The text has been updated accordingly and a thorough discussion has been added (ll.361-371).

6. Figure 4d should be replaced by images with clearer cell morphology.

Response: This assay and these representative images are intended to show differences in cell migration reflected by the absence of cells in the outlined gap rather than differences in cell morphology. Endothelial cells outside the boundaries look uniform for all groups at indicated timepoints. An enlarged image for all groups is shown below. Unfortunately, we only have brightfield images and did not collect higher magnification images or performed a staining that shows cell morphology any clearer. It remains unclear why more information on cell morphology would be needed here.

Basigin Links Altered Skeletal Stem Cell Lineage Dynamics with Glucocorticoid-induced Bone Loss and Impaired Angiogenesis (Nature Communications manuscript NCOMMS-24-47461A)

Responses to	Comments	Page
Reviewer 1	1	1
Reviewer 2	1	1
Reviewer 3	1	1-2

Changes to the text or figure legends were marked in **yellow**. No changes to Figures were made.

REVIEWER COMMENTS

Reviewer #1 (Remarks to the Author):

The question are answered appropriately and sufficiently. No more questions, and I recommend an acceptance.

Response: We thank the reviewer for the time and constructive feedback that has substantially improved our work.

Reviewer #2 (Remarks to the Author):

Thank you for addressing all of my comments. The addition of the cKO data has significantly improved the manuscript, and I have no further comments.

Response: We would like to thank the reviewer for their time and helping us to substantially improve this study.

Reviewer #3 (Remarks to the Author):

The authors have addressed most of my concerns. However, for point #2, what does it mean that CD200 is "redundant for this study"? They have defined SSCs using one single positive marker CD51 (CD45-Ter119-Tie2-CD90-6c3-CD105-CD51+, line 149-150), which is inconsistent with the mSSC definition in the Chan study (Cell, 2015, CD45-Ter119-Tie2-CD90-6c3-CD105-CD51+, reference #10). CD51 is not only expressed in mSSCs and LepR+ BMSCs, but also highly expressed in osteoprogenitors, pre-osteoblasts and mature osteoblasts. Therefore, without adding CD200 as an additional positive marker, any conclusion about SSCs in this study becomes shaky. Please avoid overclaim on SSCs throughout the manuscript if they actually analyzed a rather heterogeneous population..

Response: We thank the reviewer for the comment and wish to further clarify. While we agree that CD51 labels a broad population of cells, SSCs are additionally defined by the lack of other markers we employ to enrich for them. That means CD51-positive cells contain cell populations with different combinations of Tie2, 6C3, Thy1 and CD105 expression pattern. SSCs, however, are only found in the CD51+Tie2-6C3-Thy1-CD105- fraction. This fraction, depending on mouse age and other factors, is around 10% of the entire CD51+ cell population with our cell preparation and as seen in the presented data (**Figure A** below). Also, while we did not label with CD200 in this work, we have shown that the CD51+Tie2-6C3-Thy1-CD105- fraction of bones in adult mice is virtually entirely CD200+ (PMID: 34381212, and **Figure B** below). Therefore, we are actually looking at the CD51+Tie2-

Point-by-Point Response

6C3-Thy1-CD105-CD200+ fraction in the presented data, which remains at 10% of the entire CD51+ fraction. Since we did not label our cells with CD200 antibody, we refrained from defining them as CD200+. In the updated manuscript we have now added this information in lines 149-150 defining SSCs as CD45⁺Ter119⁻Tie2⁻CD90⁺6c3⁻CD105⁺CD51⁺[CD200⁺] and include a description as to why in the figure legend for Extended Data Figure 3a showing the flow cytometric gating strategy. We share your common concern that researchers tend to oversimplify the definition of stem cell populations by calling highly heterogeneous populations using a single positive marker “stem cells”. However, in this study we are using carefully defined SSCs based on a combination of additional negative selection markers that is based on a large body of work with functional readouts. We hope that addresses the concern and the reviewer deems our work acceptable for publication.

Figure: SSC populations within CD51+ cell fraction and CD200 expression. **A.** Representative flow cytometric gating of CD51+ cells from living, non-hematopoietic cells (PI-CD45-Ter119-) showing that CD51+ cells consist of multiple subpopulations and that the SSC fraction is around 10% of all CD51+ cells in this bone preparation. **B.** Figure panel from Ambrosi et al. 2021 *Nature* showing that compared to isotype control-stained PI-CD45-Ter119-Tie2-6C3-Thy1-CD105-CD51+ cells, CD200-FITC labeled cells are also all CD200+ in young adult (2 months) and aged (24 months) mice. Consequently, labeling for CD200 in adult mice becomes redundant when investigating SSCs. Isotype control antibody was used to stain another fraction of cells from the same sample to correctly set CD200+ cell gate.

Previous response: Thank you for the comment. We have previously shown that all SSCs in adult mice are CD200 positive (PMID: 34381212) and therefore this marker is redundant for this study. The 2015 Chan et al. study employed cells from pre-natal and early post-natal stages for functional assays. We have also demonstrated that LepR is an unspecific marker capturing both SSC and a wide spectrum of non-SSC populations (PMID: 34280086) making the marker combination used here a superior enrichment method for SSCs. Given discrepancies in protein and gene expression profiles as well as limitations in transcriptomic coverage of scRNAseq approaches, it is difficult if not impossible to identify highly enriched SSC clusters/population in 10X Genomics data. Because of that, we more broadly (carefully) assigned clusters as stem and progenitor cell enriched (SSPCs) rather than calling them SSCs. We have added detail in the updated manuscript as requested (ll.191-199).